# A novel class of *Candida glabrata* cell wall proteins with β-helix fold mediates adhesion in clinical isolates

Viktoria Reithofer[1☯], Jordan Fernández-Pereira[2☯], María Alvarado[2], Piet de Groot[2]*, Lars-Oliver Essen[1]*

1 Department of Biochemistry, Philipps-Universität, Marburg, Germany, 2 Regional Center for Biomedical Research, Castilla-La Mancha Science & Technology Park, University of Castilla–La Mancha, Albacete, Spain

☯ These authors contributed equally to this work.
* Piet.deGroot@uclm.es (PdG); essen@chemie.uni-marburg.de (LOE)

**Data Availability Statement:** All crystal structures are available from the RCSB protein structure database (accession numbers 7O9O, 7O9P, 7O9Q).

## Abstract

*Candida glabrata* is an opportunistic pathogenic yeast frequently causing infections in humans. Though it lacks typical virulence factors such as hyphal development, *C. glabrata* contains a remarkably large and diverse set of putative wall adhesins that is crucial for its success as pathogen. Here, we present an analysis of putative adhesins from the homology clusters V and VI. First, sequence similarity network analysis revealed relationships between cluster V and VI adhesins and *S. cerevisiae* haze protective factors (Hpf). Crystal structures of A-regions from cluster VI adhesins Awp1 and Awp3b reveal a parallel right-handed β-helix domain that is linked to a C-terminal β-sandwich. Structure solution of the A-region of Awp3b via single wavelength anomalous diffraction phasing revealed the largest known lanthanide cluster with 21 Gd$^{3+}$ ions. Awp1-A and Awp3b-A show structural similarity to pectate lyases but binding to neither carbohydrates nor Ca$^{2+}$ was observed. Phenotypic analysis of *awp1Δ*, *awp3Δ*, and *awp1,3Δ* double mutants did also not confirm their role as adhesins. In contrast, deletion mutants of the cluster V adhesin Awp2 in the hyperadhesive clinical isolate PEU382 demonstrated its importance for adhesion to polystyrene or glass, biofilm formation, cell aggregation and other cell surface-related phenotypes. Together with cluster III and VII adhesins our study shows that *C. glabrata* CBS138 can rely on a set of 42 Awp1-related adhesins with β-helix/α-crystallin domain architecture for modifying the surface characteristics of its cell wall.

## Author summary

Adhesion to host cells and abiotic, often hydrophobic surfaces, e.g. that of medical equipment like catheters, is an indispensable virulence factor for many pathogenic fungi. Among the latter, the yeast *Candida glabrata* excels by encoding in its genome large sets of surface-exposed cell wall proteins. Here, we show that in the clinical isolate PEU382 of *C. glabrata*, hyper-adhesiveness to plastics and the tendency to biofilm formation is

**Funding:** LOE and VR received funding from Deutsche Forschungsgemeinschaft [SFB 987]. PdG, JFP and MA received funding from the Spanish Ministry of Economy and Competitiveness (MINECO) [SAF2013-47570-P and SAF2017-86188-P], and the Regional government of Castilla-La Mancha [SBPLY/19/180501/000114 and SBPLY/19/180501/000356], the Spanish grants co-financed by the EU (FEDER). The funders had no role in study design, data collection and analysis, decision to publish, or preparation of the manuscript.

**Competing interests:** The authors have declared that no competing interests exist.

conferred by a single adhesin, Awp2. An integrative bioinformatic and structural analysis of this and the related Awp1 and Awp3 adhesins unifies four, so far separately assigned Awp clusters—III, V, VI and VII–into one consisting of 42 Awp1-related adhesins. These adhesins commonly present an N-terminal module consisting of a right-handed β-helix and an α-crystallin domain on the yeast surface and use a calcium-independent mode for adhesion. Their sheer number contrasts to the 20 members of the well characterized Epa and 7 members of the Pwp family of surface proteins. Given these findings we suggest that *C. glabrata* utilizes just two structurally distinct motifs for colonizing different host niches by adhesion: the β-helix/α-crystallin module of Awp1-related adhesins and the C-type lectin PA14-domain for Epa and Pwp proteins.

## Introduction

The yeast *Candida glabrata* is an opportunistic human pathogen that can cause mucosal, bloodstream, and medical device-related infections [1,2]. Despite a closer relationship to *Saccharomyces* than to the pathogenic CTG-clade *Candida* spp. [2], *C. glabrata* is regarded the second most common causative agent of candidiasis worldwide after *Candida albicans*. *C. glabrata* has the ability to adhere to a wide variety of biotic and abiotic surfaces [1,3], mediated by a remarkably large number of putative adhesins [4,5]. As adhesion to host tissues or to medical devices is an important first step in the establishment of fungal infections, it is regarded as an important pathogenicity factor. *C. glabrata* wall adhesins are large, modular glycosylphosphatidylinositol (GPI)-modified proteins. The N-terminal region of mature GPI cell wall proteins (GPI-CWPs)–also referred to as A-region–is believed to define the ligand-binding function. This region is followed by a low complexity region, the B-region, which is usually rich in serine and threonine residues, thus presenting abundant acceptor sites for *O*-glycosylation, and usually contains a variable number of large tandem repeats. By being linked with its processed C-terminus to cell wall β-1,6-glucans via the glycan remnant of the GPI anchor, the heavily glycosylated B-region can interact with other cell wall glycans and act as spacer molecule to present the A-region along the cell surface [1,6].

Screening the *C. glabrata* genome for genes encoding GPI-CWPs with the typical architecture of adhesins identified about 70 putative adhesins in various studies that–based on A-region sequence similarity–were categorized into seven phylogenetic clusters [5,7,8]. Well described adhesins are proteins of the Epa (epithelial adhesion) protein family (cluster I), which specify lectins that mediate adhesion to host cells by binding ligands containing terminal galactose residues [8–11]. Most proteins in the other clusters remain totally uncharacterized and their biological function or role in *C. glabrata* pathogenesis remains unknown. Unlike Epa adhesins, no orthologs of the uncharacterized Awp proteins have been described, a finding, which raises doubt about their relevance in this particular organism. However, a subset of the putative adhesins from different clusters was identified by mass spectrometric analysis of isolated cell walls in different *C. glabrata* strains and under various growth conditions. Among the identified GPI-CWPs were Epa1, 3, 6, and 7, but also a number of uncharacterized proteins from other clusters that were named Awp1–13 [4,5,12]. Their presence at the cell surface implies that these unknown putative adhesins are relevant for *C. glabrata* and support its colonization by promoting adhesion to surfaces.

This study focuses on putative *C. glabrata* adhesins from clusters V and VI. Proteomic studies identified the cluster VI protein Awp3 in walls from exponentially growing cells of reference strain *C. glabrata* CBS138/ATCC 2001, whereas Awp1 (also in cluster VI) was identified

in stationary phase cells from low-adherent strain ATCC90876 [5]. Quantitative PCR expression analysis of *AWP1* and *AWP3* in CBS138 showed higher expression of both genes in biofilm conditions compared to planktonic growth [4], however, the proteins were not identified in cell walls from CBS138 biofilm cultures, nor in hyperadhesive clinical isolates [4]. From cluster V, two GPI-CWPs that were commonly identified in isolated walls from CBS138 and hyperadhesive strains in biofilms as well as under planktonic conditions are Awp2 and Awp4. Interestingly, Awp8–11 from this same cluster have been identified only in biofilms of hyperadhesive isolates grown on polystyrene, suggesting a role for cluster V in adhesion and biofilm formation on abiotic surfaces.

Recently, an update of the *C. glabrata* CBS138 genome has been published [7]. This new genome assembly resolved several sequence errors, omissions and misassemblies in the 2004 version of the reference genome, particularly affecting (sub)telomeric regions where most of the putative adhesins are located. This also altered cluster V and VI adhesins. For instance, in the new assembly, a 5 kb insertion in the locus of *AWP3* results in two tandem genes, named Awp3a and Awp3b. Also corrected were the misassembled sequences of the highly homologous genes *AWP4* and *AWP11 (= AWP2D)*, which shifted their positions on chromosomes J and M.

Sequence analysis revealed weak similarities leading to predictions of structural similarities between the A-regions of clusters V and VI putative adhesins. The A-region of cluster V protein Awp2 also shows some similarity to the GPI-CWP Hyr1/Iff family of putative adhesins in *C. albicans*. Functional charactization studies documented that deletion of the hyphally-regulated gene *HYR1* of this family led to attenuated virulence in the mouse oral biofilm model of infection [13] and, in a different study, *IFF4* null mutants showed reduced adhesion to plastic substrate and attenuated virulence in a murine model of disseminated candidiasis [14].

In this study, we aim to shed light on the biological role of uncharacterized cluster V and VI wall adhesins in *C. glabrata*. We present an integrated study including sequence similarity network (SSN) analysis, descriptions of high-resolution three-dimensional structures of the A-regions of Awp1 and Awp3b, as well as phenotypic characterization of *AWP1-3* deletion and overexpression mutants. We show that Awp1 and Awp3b A-regions have a two domain architecture, where the N-terminal domain is structurally related to pectate lyases and bacterial β-helix domains, while the C-terminal domain adopts an α-crystallin fold. We further demonstrate that the related Awp2 GPI-CWP regulates adhesiveness to polystyrene and also controls other cell surface-related phenotypes of the hyperadhesive strain PEU382. Our study opens the path to better understand the role of these adhesin-like wall proteins in primary processes such as adhesion and biofilm formation that underlie the establishment of *C. glabrata* infections.

## Results

### Sequence similarity network of Awp1-related GPI-CWPs

Sequence similarity network analysis is a valuable tool for identifying isofunctional subfamilies within a set of related sequences. Sequences within the network are represented as "nodes", and their pairwise relationships as deduced from E-values by all-by-all BLAST analyses is depicted by lines connecting those, referred to as "edges". This procedure leads to the formation of clusters of nodes that represent protein subfamilies [15]. We generated a SSN using the A-regions of the previously identified *C. glabrata* GPI-CWPs Awp1-Awp4 from cluster VI as seeds for identifying related orthologs by extensive PSI-BLAST searches. Interestingly, the resulting SSN of the Awp1/Awp3 orthologs (Fig 1) is dominated by bacterial species (93.0%; 9569/10290) whereas only 6.9% belong to fungi and here exclusively to ascomycetes (711/

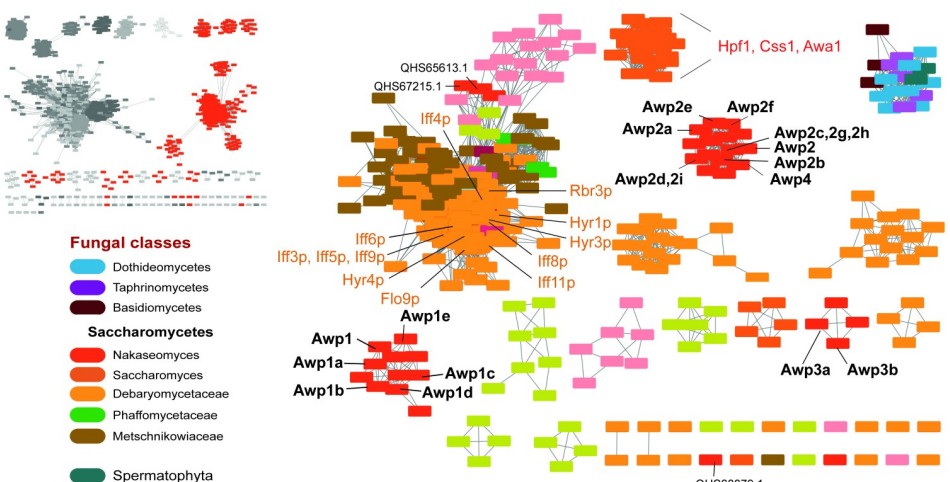

**Fig 1. SSN analysis of the class of fungal cell wall proteins of the Awp1/Hyr1/Hpf1-type.** Upper left, the SSN of orthologs of Awp1-Awp4 is shown (E-value cut-off $10^{-20}$), which have been accumulated by ten repeated rounds of PSI-BLAST for each before merging the results and removing redundancy prior to SSN-analysis. The Interpro family of 'N-terminal, hyphally-regulated cell wall proteins' (IPR021031) is a subset of this SSN. Each of the 4507 nodes (left, E-value cutoff $10^{-20}$) covers sequences with pairwise sequence identities of at least 80% (total: 11737 sequences), 445 nodes (711 sequences) are of fungal (red), the remainder (grey) of bacterial origin. Right, the majority of fungal orthologs are found in Saccharomycetes (E-value cut-off $10^{-25}$). The structurally characterized A-regions of Awp1 and Awp3b fall in two separate *C. glabrata* subclusters besides the Iff4 subcluster. In *C. albicans*, at least twelve paralogs of the same domain type can be identified (orange labels), in *S. cerevisiae* strains Awa1, Hpf1 and Css1 form a common subcluster. The nodes are colored by fungal clade, with red being *C. glabrata* (Nakaseomyces); light orange designated the CTG clade, including *Candida albicans*.

10290). The latter contain as a subset the INTERPRO protein family IPR021031, which covers conserved N-terminal domains of fungal proteins like Hpf1 from *Saccharomyces cerevisiae* and Hyr1 from *Candida albicans*. Hyr1 is apparently involved in hyphal formation and promotion of biofilm formation. In contrast, the former protein, Hpf1, is a heavily glycosylated manno-protein with still unknown biological function for *S. cerevisiae*. Its assignment as a haze-protective factor was solely derived from the oenologically useful trait during white wine production [16]. Another difference between them is the presence of at least 12 Hyr1 paralogs in *C. albicans* (Fig 1), named as Iff3-Iff6, Iff8-Iff9, Iff11, Iff14, Flo9 besides Hyr3 and Hyr4, whereas *S. cerevisiae* S288c harbors only two, Hpf1 and Css1.

In *Candida glabrata* CBS138 we found 22 paralogs of Awp1 (S1 Fig), which exceeds even the number of Epa1-like adhesins with 20 paralogs [7]. Accordingly, we simplify the nomenclature for Awp1-related GPI-CWPs in *C. glabrata* as follows: eight paralogs (Awp1, Awp1a-Awp1e, Awp3a, and Awp3b) belong to the previously annotated cluster VI. Thirteen form cluster V with ten of them (Awp2a-Awp2i, Awp4) being closely related to Awp2 (pairwise sequence identities 54% - 95%). Consequently, former Awp8-Awp11 were reassigned to Awp2a, Awp2b, Awp2c, Awp2d, respectively. The remaining two cluster V members (Genbank IDs: QHS67215.1, QHS65613.1) share more similarities with *C. albicans* proteins (Fig 1). One paralog (QHS68879.1) has not been assigned before to cluster V or VI and lacks any predicted disulfide bridges for its A-region (see below).

## Awp2 but not Awp1 or Awp3 governs adhesion to polystyrene

The importance of Awp1-related GPI-CWPs for fungal adhesion and other surface characteristics was studied by generating deletion mutants for *AWP1*, *AWP2* and *AWP3* using sequence information from genome assembly 2 in the *Candida* Genome Database. So far, Awp1 and

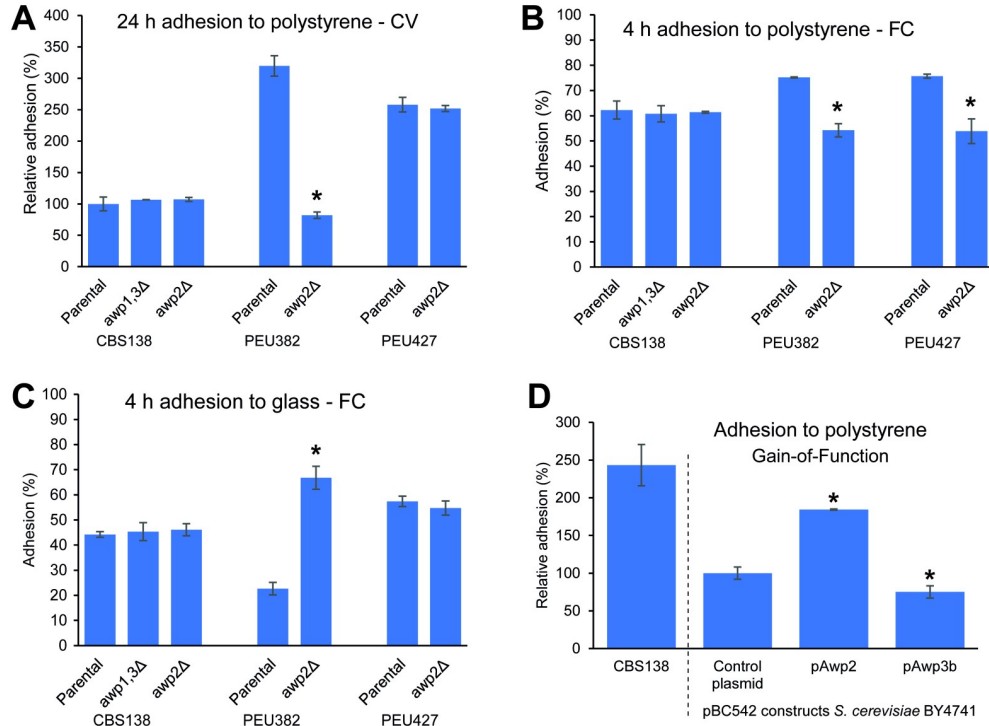

**Fig 2. Awp2 mediates adhesion to plastic.** (A) Adhesion to polystyrene after 24 h measured by Crystal violet (CV) staining. (B) Adhesion to polystyrene after 4 h measured by FC. (C) Adhesion to glass being measured after 4 h by flow cytometry (FC). Data in (A) are normalized to CBS138. *AWP2* was deleted in three genetic backgrounds: reference strain CBS138, and hyperadhesive strains PEU382 and PEU427. *AWP1* and *AWP3* were deleted only in CBS138 as the corresponding proteins have not been identified in the hyperadhesive strains. (D) Adhesion to polystyrene after 24 h of gain-of-function mutants in *S. cerevisiae*, normalized to parental strain BY4741. Asterisks indicate significant differences of mutants compared to their parental strains (A) and (B) or the empty plasmid control (D).

Awp3 are the only two cluster VI GPI-CWPs identified in cell wall preparations. A recent update of the *C. glabrata* genome [7] indicated that our *awp3Δ* mutants in fact lack the whole gene tandem *AWP3a - AWP3b* (S2 Fig). As neither the *awp1Δ* nor the *awp3Δ* mutants showed any phenotypes in preliminary assays of fitness, adhesion, aggregation, and drug sensitivities, we also created a double mutant where the tandem *AWP3a/AWP3b* was deleted in the *awp1Δ* background and used this mutant for the studies described below. Of cluster V, two proteins that are often encountered in cell wall preparations are Awp2 and Awp4, which are part of the same SSN cluster (Fig 1). As the *AWP4* sequence in genome assembly 2 contains various errors leading to frameshifts, we focused on *AWP2* and deleted this gene in three genetic backgrounds: in reference strain CBS138 as well as in the hyperadherent clinical strains PEU382 and PEU427.

Mutants of Awp1-related GPI-CWPs were first analyzed for their capacity to adhere to polystyrene (Fig 2A). Upon 24 h of incubation, adhesion to polystyrene was clearly diminished by deleting *AWP2* in PEU382, a strain with strong hyperadherence to plastic, resulting in a low level of adherence resembling that of reference strain CBS138. In CBS138 or PEU427, however, no lowering in adherence to polystyrene was observed upon deleting this gene. One could hypothesize that the effect of loss of Awp2 might be obscured by functional redundancy with other cluster V adhesins, and this may be particularly relevant in PEU427, where the related cluster V proteins Awp2a, Awp2b, and Awp2c (previous annotations: Awp8, 9 and 10) have been identified in cell walls of 24 h biofilms [12] (S2 Table). Therefore, we tested

adherence to polystyrene after shorter incubation for 4 h in PBS to avoid extensive biofilm formation (Fig 2B). In this case, in both hyperadherent strains PEU382 and PEU427, deletion of Awp2 clearly diminished adherence to a level similar to the reference strain. This Awp2-mediated adhesion of *C. glabrata* to polystyrene is independent of calcium and other bivalent cations, as addition of EDTA to the assays caused no loss of strain-specific adhesion (S3 Fig). A dominant role of Awp2 for causing the hydrophobic surface character of the PEU382 strain can also be inferred from the twofold diminished adhesion to hydrophilic glass surfaces by PEU382 when compared to CBS138 and PEU427 (Fig 2C). Deletion of *AWP2* in PEU382 restores the adhesion to glass surfaces to even higher levels than exercised by CBS138 and PEU427, indicating that at least some Awp1-related A-regions have anti-adhesive functions as well.

We then tested the mutants´ adhesiveness to human Hela and HaCaT cells but found no difference with their respective parental strains (Table 1). This implies no role of Awp2 in binding of these *C. glabrata* strains to human cells. For the cluster VI *awp1,3Δ* mutants, no phenotypes in adhesion were detected that could confirm their role as adhesins. Interestingly, the hyperadhesive PEU427 strain is the only strain that mediates adhesion to keratinocyte-like HaCaT cells at a similar level as found for the adhesion of all examined *C. glabrata* strains against cervical cancer-derived HeLa cells. Accordingly, the PEU427 strain may have a unique set of cell surface adhesins, that has been evolved towards the specific interaction with the epidermis.

To further demonstrate the involvement of Awp2 in adhesion to polystyrene, hybrid gain-of-function constructs, containing the A-regions of Awp2 and Awp3b in front of the Epa1 Ser/Thr-rich region [8], were expressed at the cell surface of a non-adhesive *S. cerevisiae* strain (Fig 2D). Consistent with the results obtained with the deletion mutants, expression of Awp2 but not of Awp3 paralogs in *S. cerevisiae* led to a clear, almost two-fold, increase of adhesiveness to polystyrene during a 24 h incubation.

We then checked the *AWP* deletion mutants for a range of other phenotypes including fitness, sensitivity to cell wall-perturbing agents like calcofluor white (CFW) and SDS, antifungals like amphotericin B, flucanozole, isavuconazole, caspofungin, and micafungin, sedimentation, aggregation, surface hydrophobicity, and contents of cell wall components like chitin and protein (Figs 3 and S4). Compared to reference strain CBS138, the hyperadhesive strain PEU382 has been documented to have lower growth rates, increased resistance to the β-1,3-glucan hydrolyzing enzyme zymolyase and to the cell wall perturbant CFW, fast sedimentation, and hyperaggregation, among other phenotypes [12]. All these PEU382 phenotypes were completely eliminated by deleting *awp2Δ* in this background (Fig 3A–3D and 3F). Interestingly, the hyperaggregation and sedimentation of parental PEU382 cells grown to stationary

**Table 1. Adhesion of *C. glabrata* awp mutants and parental strains to human HeLa and HaCaT cells.**

| *C. glabrata* strain | | HeLa (%) | HaCaT (%) |
|---|---|---|---|
| CBS138 | Parental | 100 ± 4.2[a] | 18.2 ± 1.4 |
| | *awp1,3Δ* | 103.7 ± 17.4 | 19.9 ± 2.4 |
| | *awp2Δ* | 100.1 ± 9.5 | 18.9 ± 1.7 |
| PEU382 | Parental | 103.7 ± 5.4 | 20.5 ± 2.5 |
| | *awp2Δ* | 122.8 ± 17.0 | 21.9 ± 2.0 |
| PEU427 | Parental | 152.7 ± 11.2 | 92.2 ± 11.6 |
| | *awp2Δ* | 158.8 ± 14.3 | 94.7 ± 6.6 |

[a] All data in the table is normalized to CBS138 adhesion to Hela cells showing an absolute adhesion level of 31%

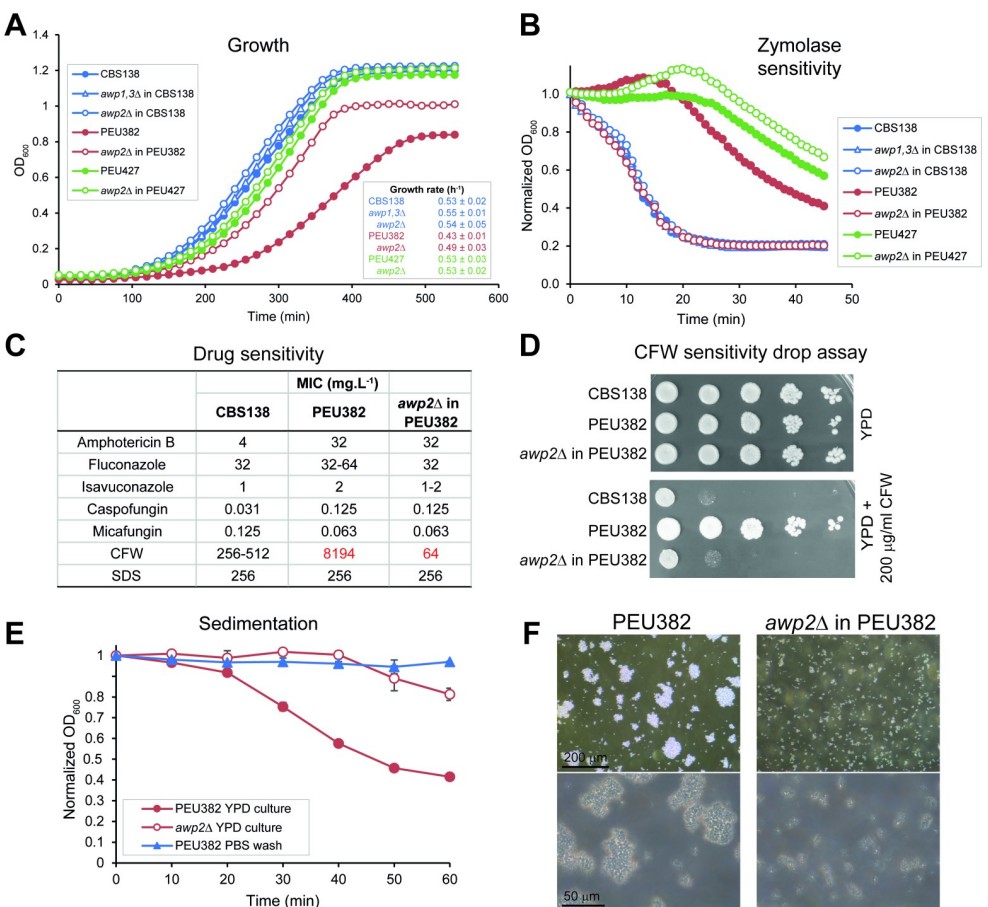

**Fig 3. Deletion of *AWP2* overrides cell surface-related phenotypes of hyperadhesive strain PEU382.** (A) Growth rate, (B) zymolyase sensitivity, (C) drug sensitivities using microbroth dilution methods, (D) Calcofluor white drop assay, (E) sedimentation and (F) aggregation of *awp* mutants. For reasons of clarity (C-F) are limited to CBS138, PEU382, and *awp2Δ* in PEU382.

phase in YPD was abolished when the cells were washed with PBS (Fig 3E). This also occurred by adding PBS to stationary phase cultures in YPD prior to the sedimentation test (S5A Fig) but not when NaOH was added to reach the same pH of 6.5. Interestingly, the same effect could be achieved by adding fresh YPD media, but not by EDTA or YPD media conditioned before by *C. glabrata* growth therein (S5B and S5C Fig). These results suggest that there may other factors than phosphate that may inhibit hyperaggregation of PEU382. Nevertheless, colorimetric assay measurements of protein and chitin content of cell walls isolated from planktonic (exponential phase) and 24 h biofilm cells (S4 Fig) did not reveal drastic changes between deletion mutants and parental strains. The increased protein (planktonic) and chitin (biofilms) contents observed in PEU382 compared to CBS138 were diminished by deletion of *AWP2* in PEU382. In drug sensitivity assays (antifungals, SDS, S3 Table) no differences between strains PEU382 and CBS138 were observed and deletion of *AWP2* in PEU382 did not result in any phenotype. Altogether, these data underline the importance of Awp2 in PEU382, not only for hyperadherence but also for other altered cell surface characteristics that make this clinical isolate so atypical.

Contrasting with the many surface phenotypes of *awp2Δ* in PEU382, *awp2Δ* mutants in PEU427 and CBS138 as well as *awp1,3Δ* double mutants in CBS138 lacked obvious cell

surface-related phenotypes as only minor alterations were observed. In detail, the increased zymolyase resistance of PEU427 compared to CBS138 was not abolished by deletion of *AWP2* as happened in PEU382 background but instead became slightly more pronounced. In CBS138 background, *awp2Δ* mutants showed slight alterations in surface hydrophobicity in stationary phase cells and cell wall protein content in both planktonic as well as biofilm cells. For the *awp1,3Δ* mutants, the only observed alterations were lower protein and higher chitin contents in biofilm cells.

## Structures of Awp1-related A-regions

For structural studies, we overproduced the A-regions of Awp1 and Awp3b in *E. coli* and readily obtained crystals for both of them. In contrast, the recombinant Awp2 A-region was found to form only insoluble inclusion bodies. Crystals of the Awp3b A-region (Awp3b-A) belong to rhombohedral space group *R*32 with one molecule per asymmetric unit. As no protein structure with sufficient sequence identity for molecular replacement (MR) was available in the Protein Databank (PDB), we determined phases by single wavelength anomalous diffraction (SAD). Awp3b A-region crystals soaked with Gd(OAc)$_3$ diffracted to a resolution of 2 Å, therefrom derived SAD phases were used to solve a native dataset of 1.55 Å resolution. Only few regions with uninterpretable electron density are contained in the Awp3b-A•Gd$^{3+}$-complex, namely the loops consisting of residues S75–D82 and S320–E322. However, these regions are clearly defined in the Awp3b-A structure. Data collection and refinement statistics are summarized in Table 2. The conformations of some loops, especially T1 (see below), differ in the Awp3b A-region structures and hence indicate some flexibility for these regions.

The Awp1-A crystals belong to space group *P*4$_3$2$_1$2 with two molecules per asymmetric unit and diffracted to a resolution of 1.85 Å. The Awp1-A structure was solved by MR using the Awp3b-A structure as template, followed by 20 cycles of model building in ARP/wARP. Structural similarity between the A-regions of Awp1 and Awp3b is evident from a calculated root mean square deviation (RMSD) of 1.36 Å for 223 Cα positions. This is consistent with the notion that both proteins belong to the same cluster of putative *C. glabrata* adhesins as they share a sequence identity of 25.1% for their A-regions (Awp1: S18-A324; Awp3b: D20-E344).

The A-regions of Awp1 and Awp3b (Fig 4A) consist of 34 and 33 β-strands, respectively, which form a two-domain architecture with a parallel right-handed β-helix at its N-terminus (Awp1: S18-V240 with strands β0-β26, Awp3b: D20-I254 with β1-β26) that is followed by a β-sandwich domain with α-crystallin fold (Awp1: V241-Q314, Awp3b: V255-E334; both with strands β27- β33). The C-termini of the A-regions are covalently attached to the α-crystallin domain via a disulfide bridge to a cysteine residue in the loop linking β-strands 31 and 32 (Figs 4A and 5B; Awp1: C284-C322, Awp3b: C304-C341). Disulfide-bridging between A-domain cores and their C-termini has been observed before in fungal adhesins with PA14 [6,9,17] and fibronectin-type domains [18].

Due to the lack of an experimental structure, we subjected the sequence of the Awp2 A-region to template-independent structure prediction [19] by transformer-restrained Rosetta (trRosetta). The structural model of Awp2 (Fig 4B) is based on 195 homologous sequences and shows the same fold as the Awp1 and Awp3b A-regions with r.m.s.d. values of 2.60 and 2.68 Å for 223/239 Cα positions, respectively. These modest deviations are comparable to differences between the experimental structures of Awp1 and Awp3b and their corresponding trRosetta models (3.39 Å for 216 and 1.52 Å for 198 Cα positions, Fig 5A). Like other cluster V adhesins, Awp2 is predicted to form other disulfide bridges (Fig 5C) than the cluster VI adhesins by linking the C-terminal cysteine residues of the CxxC motif subsequent to its A-region to cysteines in β-helix turns 7 and 8 (Awp2: C327-C201, C330-C184; S1 and S6 Figs). As a consequence the

**Table 2. Data collection and refinement statistics for crystal structures of Awp1- and Awp3b-A regions.** Deposition numbers for the RCSB protein data bank are given in parentheses. Data collection statistics is generated by SCALA [65].

| | Awp1-A | Awp3b-A | Awp3b-A•Gd$^{3+}$ |
|---|---|---|---|
| | (7O9Q) | (7O9O) | (7O9P) |
| Data collection | | | |
| X-ray source | ESRF, ID29 | ESRF, ID23-1 | ESRF, ID29 |
| Wavelength | 0.97717 | 0.97625 | 1.71237 |
| Space group | $P\,4_3\,2_1\,2$ | $R\,3\,2$ | $R\,3\,2$ |
| Unit cell parameters (Å) | $a = b = 83.28$, $c = 274.24$ | $a = b = 147.97$, $c = 117.77$ | $a = b = 144.4$, $c = 113.95$ |
| Resolution range (Å)[a] | 45.81–1.85 (1.92–1.85) | 53.51–1.55 (1.61–1.55) | 27.41–1.99 (2.06–1.99) |
| Total No. of reflections[a] | 165589 (16103) | 134731 (13321) | 62641 (6200) |
| No. of unique relfections[a] | 83156 (8101) | 69278 (6889) | 31321 (3100) |
| $R_{merge}$ (%)[a] | 2.97 (34.94) | 3.627 (42.99) | 3.672 (12.86) |
| $I/\sigma(I)$[a] | 12.64 (1.62) | 10.68 (1.84) | 18.42 (4.90) |
| Completeness (%)[a] | 99.16 (97.48) | 96.96 (97.30) | 99.92 (100.00) |
| Multiplicity[a] | 2.0 (2.0) | 1.9 (1.9) | 2.0 (2.0) |
| $CC_{1/2}$[a] | 0.998 (0.95) | 0.999 (0.431) | 0.997 (0.924) |
| Refinement | | | |
| $R_{work}/R_{free}$ (%) | 18.79/20.83 | 15.93/18.79 | 19.03/22.78 |
| No. of atoms | 5262 | 3117 | 2658 |
| Average B factor (Å$^2$) | 51.43 | 28.78 | 37.13 |
| R. m. s. deviations | | | |
| Bond length (Å$^2$) | 0.004 | 0.008 | 0.014 |
| Bond angles (°) | 0.67 | 0.99 | 2.02 |
| Clash score[b] | 0.99 | 2.02 | 3.59 |
| Ramachandran plot (%)[b] | | | |
| Favoured[b] | 96.89 | 96.93 | 96.68 |
| Allowed[b] | 2.62 | 3.07 | 2.99 |
| Outliers[b] | 0.49 | 0.00 | 0.33 |
| Rotamer outliers (%)[b] | 1.31 | 0.35 | 3.09 |

[a] Values for the outer resolution shell are given in parentheses.

[b] Quality values as provided by MolProbity.

C-termini of the rod-like A-regions of cluster V adhesins are placed on the central part of the A-regions rather than at their tip as found for Awp1 and Awp3b. Given that these A-regions are followed by long, heavily *O*-glycosylated repeats of the B-region, cluster-specific differences may cause different orientations of the A-regions along the surface of the *C. glabrata* cell wall.

According to the nomenclature for β-helices by Yoder & Jurnak [20], the three β-strands forming a single turn in β-helices are referred to as PB1, PB2, and PB3 (Fig 4C); loops between them are labeled T1 (connecting PB1 and PB2), T2 (PB2 and PB3), and T3 (PB3 and PB1 of next turn). Additionally, Awp1-A has a truncated turn comprising only PB3-T3 at the N-terminus (β0: S18-P24). As right-handed β-helices, the T2 and T3 loops of Awp1-A and Awp3b-A are very short, whereas T1 loops are more extended. Interestingly, the T1-loops of β-helix turns 4–6 from Awp3b-A are cross-linked by two disulfide bridges, C115-C145 and C144-C178 (Fig 4A). These features, which were found to stabilize analogous loop conformations in Epa-like adhesion domains of *C. glabrata* [9], are lacking in Awp1 and other Awp1-related GPI-CWPs (S1 Fig).

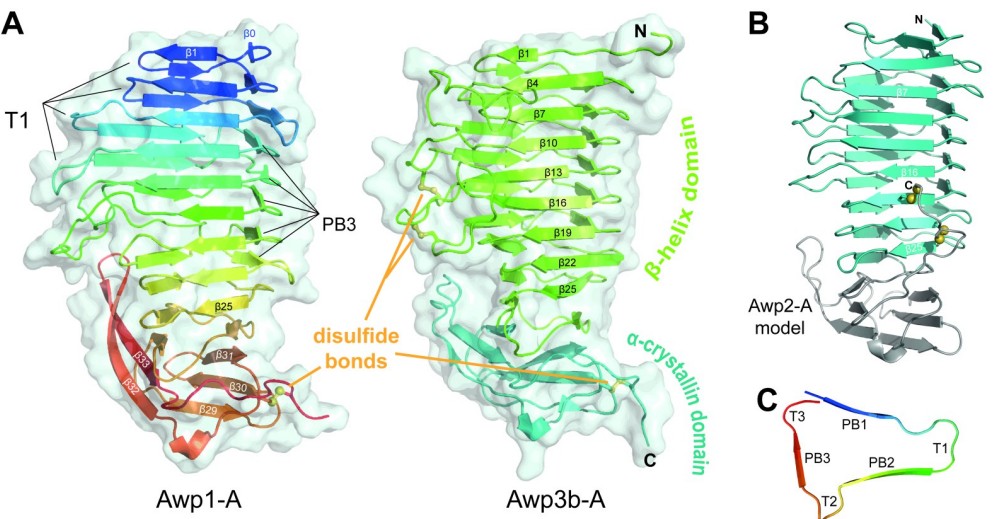

**Fig 4. Structures of the Awp1, Awp2 and Awp3b A-regions.** (A) Left, crystal structure of the Awp1 A-region. The nine β-helix turns are colored according to S1 Fig. The C-terminal α-crystallin domain is depicted in orange. Right, crystal structure of the Awp3b A-region. (B) Model of the Awp2 A-region based on transformer-restrained modelling by trRosetta. (C) Architecture and nomenclature of β-helix turns.

## The two-domain architecture of Awp1-related A-regions

A feature often found in β-helices is the repetitive stacking of distinct residue types along the outer and inner sides of the β-helix [21]. These stacks are regarded to provide additional stability and rigidity to the structures of β-helix proteins. On the inner sides, Awp1-A and Awp3b-A exhibit stacks of hydrophobic amino acids placed on all three sheets composing the β-helix, mainly Val, Leu, Ile, and Phe, and occasionally Tyr (Fig 6A and 6B). The twisted conformation of the β-helix results in a slight offset between following consecutive residues of the stack, thereby preventing unfavourable alignment of aromatic side chains due to overlap of

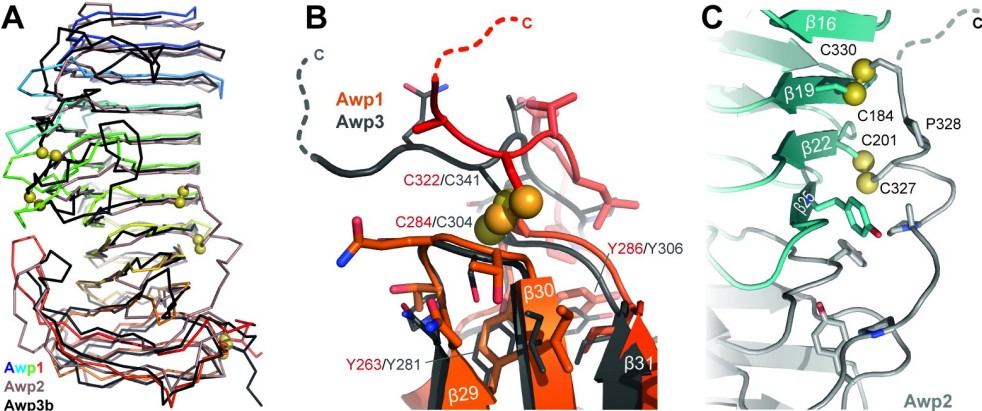

**Fig 5. Similarity and disulfide bridges in Awp1, Awp2 and Awp3b A-regions.** (A) Superpositions of the Awp1 and Awp3b A-region crystal structures with the trRosetta model of Awp2A. Despite the limited sequence identity of 24% to the Awp3b A-region, the Awp3b-A structure closely resembles the Awp1 A-region with an r.m.s.d. of 1.4 Å for 223 Cα-atoms. (B) Disulfide bridges between the C-terminal cysteine residue and the α-crystallin domains of Awp1 and Awp3b A-regions. (C) Predicted disulfide bridges between the C-terminus and β-helix turns of the trRosetta model of the Awp2 A-region.

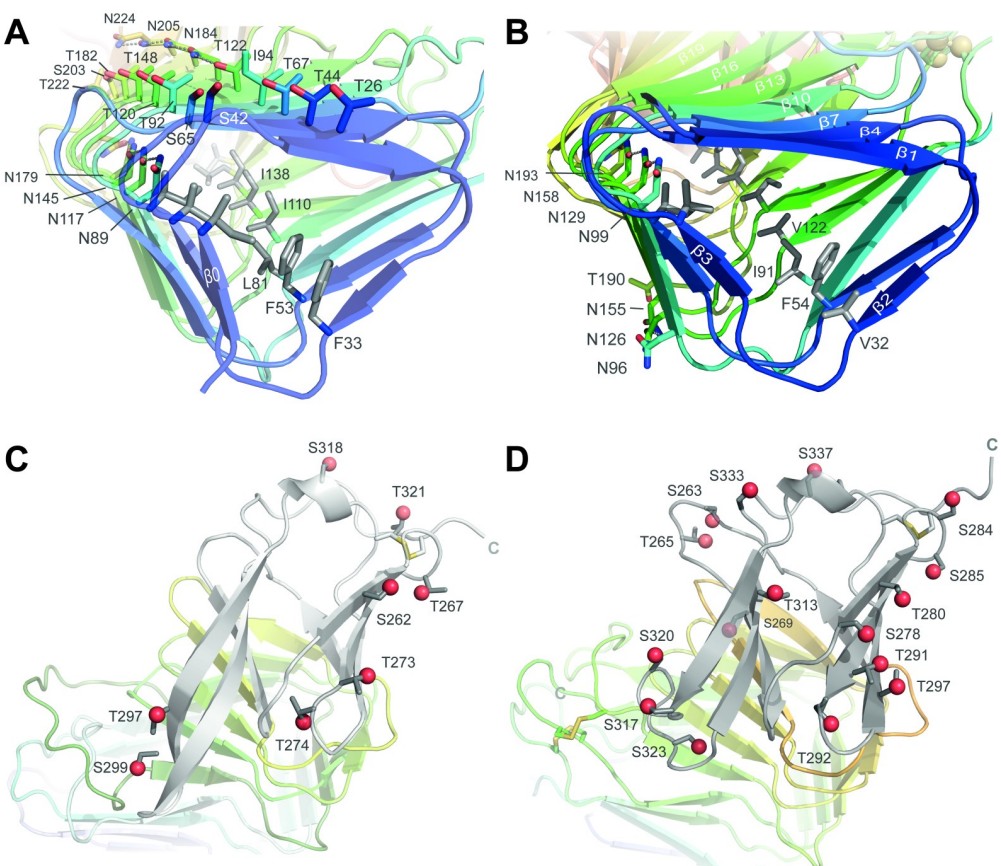

**Fig 6. Stacking patterns and *O*-glycosylation sites within the Awp1 and Awp3b A-regions.** Top, view along the β-helix domains showing the repetitive bands of hydrophobic (gray) and serine, threonine and asparagine residues (colored) along the inner and outer β-helix areas of Awp1 (A) and Awp3b (B) A-regions. Bottom, *O*-glycosylation sites as predicted by NetOGlyc 4.0 for the A-regions of Awp1 (C) and Awp3b (D). The hydroxyl groups to be covalently modified by a fungal oligomannan are highlighted as spheres. Only eight of nine predicted *O*-glycosylation sites in Awp1 (Awp3b: 16 of 17) are depicted because S254 (Awp3b: T268) is buried within the α-crystallin domain (gray) and hence unaccessible for *O*-glycosylation. Notably, no or only a single *O*-glycosylation site was predicted for both A-regions in the β-helix domain.

π-electron systems. β-helices of Awp1-A and Awp3b-A are characterized by highly conserved internal Asn ladders (Awp1: N89, N117, N145, N179; Awp3b: N99, N129, N158, N193; Awp2 model: N103, N129, N152, N179), whose asparagines adopt trans/gauche⁻ rotamers ($\chi_1$ ~-179°, $\chi_2$ ~44°) and are found as last residues of PB3 strands starting from β-helix turn 4. These internal Asn ladders are capped by hydrogen-bonds to a tilted asparagine or glutamine residue (Awp1: N221, Awp2: Q220, Awp3b: Q234) that follows the PB3 strand of helix turn 8. Another feature of striking regularity can be observed along the PB1 sheet of the Awp1-A β-helix. Here, a Ser/Thr ladder stretches over 8 β-turns along the exterior face at the beginning of each β-strand (Fig 6A; S42, S65, T92, T120, T148, T182, S203, and T222). In a +2 position, a second outer stack consisting of four Thr and four Asn, interrupted by an isoleucine residue, stretches along the whole PB1 sheet. The four asparagine side chains ($\chi_1$ ~-63°, $\chi_2$ ~-53°) resemble the interior Asn stack by forming a hydrogen bonding network between the side chains' carboxamide groups. Stacking on the exterior side of β-helix sheets has been described in several proteins, but seems to play a role in providing a hydrophobic environment for

packing of other structural features (e. g. an α-helix) towards the β-helix in many cases [21], which is not the case for Awp1-A.

Serine and threonine residues are known to be potential target sites for *O*-linked glycosylation, whereas asparagines within the sequence motif N-X-(S/T) may be subjected to *N*-linked glycosylation. Prediction of *O*-glycosylation sites in Awp1-related adhesins was done using NetOGlyc 4.0 [22] and identified numerous potential glycosylation sites, especially in the B-regions, where one third, on average, of the residues are predicted to carry a covalently attached oligomannan modification. (S4 Table). This coincides with glycosylation predictions performed on B-regions of many other fungal cell wall proteins [23]. Interestingly, the modification pattern is diverse for the A-region of the Awp1-related adhesins. The A-regions of the Awp1 and Awp3 subclusters (Fig 1) are almost all predicted to be heavily *O*-glycosylated at the α-crystallin domain and the following cysteine-comprising stretch (Awp1: 8 sites; Awp3b: 16 Sites), but not along the β-helix domain (Fig 6C and 6D). Accordingly, none of the potentially *O*-glycosylated residues is involved in formation of the Ser/Thr ladder at the PB1 sheet of the Awp1-A β-helix domain. The roles of the Ser/Thr and the Asn stacks on the surfaces Awp1-related A-domain β-helices apart from structural stabilization of the β-helical turns [24] hence remain elusive. In contrast to the Awp1 and Awp3 subclusters, members of the Awp2 subcluster, i.e. Awp2, Awp2a-Awp2i and Awp4, apparently lack *O*-glycosylation sites on their α-crystalline domains (S4 Table). Here, the only exceptions are Awp2d with five and Awp2i with two predicted sites. Obviously, one cannot decide whether this different modification pattern is caused either by the alternative attachment mode of Awp2 A-regions to the B-region via the CxxC motif that follows the A-region or the sheer size of their B-regions, on average ~2000 residues (Awp1 subcluster: ~1000; Awp3 subcluster: ~850).

*N*-linked glycosylation sites as predicted by NetNGlyc 1.0 [25] are only scarcely found in Awp1-related adhesins (S4 Table). In Awp1, N224, the most C-terminal residue of the Thr/Asn-stack on the PB1 sheet, may be such a potential target site for *N*-linked glycosylation, but this site is not predicted for other A-regions of Awp1-related adhesins. Interestingly, in the α-crystallin domains of Awp2-subcluster members a conserved site for *N*-linked glycosylation can be found. In Awp2, the N291 site resides on the loop linking β-strands 31 and 32; in Awp1 and Awp3 subcluster adhesins, this loop is used for linking the C-terminal end of the A-region to the α-crystallin domain by a disulfide bridge (Fig 5B).

Other surface cues that may play a role in the adhesive/anti-adhesive properties of Awp1-related adhesins are surface electrostatics and hydrophobicity. In terms of electrostatics, the Awp3b A-region is clearly distinct from Awp1 and Awp2 due the former's negatively charged surface (Fig 7A). This charge distribution of the Awp3b A-region is apparently the reason for its capability to bind *in crystallo* 42 $Gd^{3+}$ ions on its surface, with 21 of them forming the so far largest structurally defined lanthanide cluster (Fig 7B). Only five $Gd^{3+}$ ions are found to coordinate to the C-terminal α-crystallin domain. This propensity for coordinating multivalent cations may modulate any adhesive/anti-adhesive effects caused by Awp3b. Hydrophobic surface bands made up of aromatic residues have been described for the Flo11 adhesins of *Saccharomyces cerevisiae* [18,26], which mediate, similar to Awp2 of *C. glabrata*, the adhesiveness to plastic surfaces. Interestingly, groups of surface-exposed tyrosine side chains can be delineated only in the model of the Awp2 A-region (S7 Fig) with overall 12 residues, whereas the crystal structures of Awp1 and Awp3b show only 4 and 6 tyrosines on the protein surfaces, respectively.

## Structural relationship to other β-helix proteins

Structural similarity of Awp1-A and Awp3b-A to proteins deposited in the PDB was analyzed by pairwise 3D alignments with PDBeFold v2.59 with the default cut-off of 70% for lowest

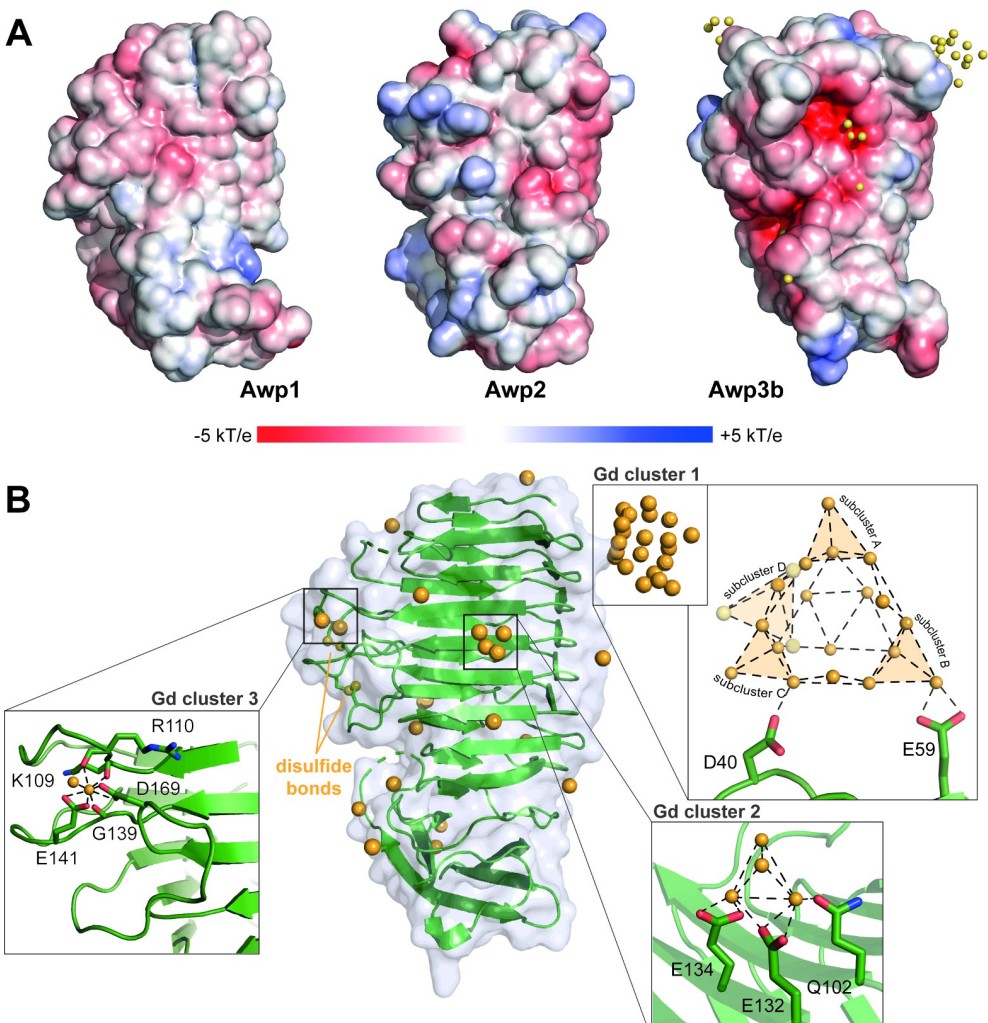

**Fig 7. Electrostatics and lanthanide binding by Awp1, Awp2, and Awp3b A-regions.** (A) Electrostatic molecular surface potentials of Awp1, Awp2, and Awp3b. The Awp3b surface structure is shown in superposition to the $Gd^{3+}$ binding sites of the Awp3b•Gd complex. (B) The 42 $Gd^{3+}$ binding sites of the Awp3b•Gd complex. Three $Gd^{3+}$ clusters are found on the surface of the β-helix domain, comprising either 21, 4 or two lanthanide ions. Cluster 1 with its 21 $Gd^{3+}$ ions is coordinated by only two residues, D40 and E59. It contains four tetrahedral subclusters (A, B, C, and D); subclusters A, B, and C with distances of 3.3–4.1 Å between their $Gd^{3+}$ ions are connected to each other by triangular planar clusters composed of three $Gd^{3+}$, with which they form a basket-like shape with three-fold symmetry. Distances of ions participating in composing those triangles range from 3.5 to 4.4 Å. Subcluster D is associated to the basket-like shape via a single $Gd^{3+}$ ion. Atoms in this subcluster are a bit further apart from each other when compared to other tetrahedral clusters, namely 3.7–4.4 Å. Cluster 2 with four $Gd^{3+}$ ions forms a regular tetrahedron that is connected to the protein via Q102, E132 and E134. Distances between the $Gd^{3+}$ ions range from 2.8 to 3.8 Å, and the $Gd^{3+}$ ions are 2.4 Å away from the carboxyl group O of the coordinating residues. Tetrahedral geometries of lanthanide clusters have been reported previously for complexes of the *S. cerevisiae* Flo5 A-domain with $Gd^{3+}$[38]. Such clusters are of high interest for the development of novel contrast agents in magnetic resonance imaging [72].

acceptable similarity [27]. A number of proteins was identified to be similar to the β-helices of these putative adhesins, including the heme-haemopexin binding HxuA from *Haemophilus influenza* [28], a variety of polysaccharide lyases from different organisms (e. g. the pectate lyase Bsp165PelA from *Bacillus* sp. N16-5 [29], pectate lyase A from *Erwinia chrysanthemi* [30], alginate lyase from *Paenibacillus* sp. str. FPU-7 [31], as well as other polysaccharide-binding proteins (e. g. the chitin-binding polysaccharide lyase-like protein Cthe_2159 from *C.*

*thermocellum* [32], the Vi-antigen lyase VexL from *Achromobacter denitrificans* [33] or the ser-ine-rich repeat protein (SRRP$_{100-23}$) from *Lactobacillus reuteri* [34]. These proteins all contain a three-faced right-handed β-helix. In general, sequence conservation was observed to be low, with sequence identities between Awp3b-A and search results ranging from 4.3% to 14.8% and RMSD values ranging from 2.63 to 6.04 Å, which indicates structural similarity. Similar results have also been observed for other β-helix proteins [28,32]. Interestingly, characteristic features of β-helix proteins such as the vWiDH sequence motif are not observed in the Awp1-A and Awp3b-A β-helices. Furthermore, many of the other β-helix proteins contain extensively long T3 loop regions, which often form additional secondary structure elements packed against the β-helix [21]. Such extrahelical domains are not seen in Awp1-related A-regions where the T1 loops represent the longest loops and do not form any regular structures.

Parallel β-helices have been identified in the polysaccharide lyase families PL1, PL3, PL6, and PL9 [35]. In those enzymes, as well as in the polysaccharide lyase-like Cthe_2159 encoun-tered in the PDBeFold search, $Ca^{2+}$ is required for ligand recognition [32,35]. $Ca^{2+}$ depen-dency has also been observed for ligand binding in the Epa family of *C. glabrata* adhesins [36]. The use of lanthanides as probes for $Ca^{2+}$ binding sites has been described on several occasions including the C-type lectin adhesion domain of the yeast adhesin Flo5 [37,38]. Consistent with this, potential $Ca^{2+}$ coordination sites in Awp3b-A were delineated by numerous $Ca^{2+}$ mim-icking $Gd^{3+}$ ions in the structure of Awp3b-A-Gd. Most residues liganding to $Gd^{3+}$ ions like D40 and E59, which coordinate the $Gd_{21}$-cluster 1, or Q102, E132 and E134, which bind to the tetranuclear $Gd^{3+}$ cluster 2 (Fig 7B) are not conserved among Awp1-related A-regions. An analysis of the Awp3b-A•$Gd^{3+}$ complex by the CMM server [39] shows that only the binding site for $Gd^{3+}$ cluster 3, which involves the side chains of E141, D169 and the carbonyl groups of K109 and R110, fulfills the geometric criteria expected for a mononuclear $Ca^{2+}$-binding site. Accordingly, a structural alignment with the pectate lyase C from *Dickeya chrysanthemi* (PDB: 2EWE) as a representative of the search results from the PDBeFold search indicates that none of these putative $Ca^{2+}$ binding sites in Awp3b-A is located at positions equivalent to $Ca^{2+}$ bind-ing sites in the active site of pectate lyases and pectate lyase-related enzymes. Furthermore, there is no $Ca^{2+}$ binding site predicted for Awp1-A because none of the lanthanides from the crystallization condition was observed in the Awp1 A-region structure. Despite their structural similarity to the aforementioned glycan-processing enzymes, our recombinant Awp1 and Awp3b A-regions failed to bind to any glycans as covered by the mammalian glycan array, ver-sion 5.2, from the Consortium of Functional Glycomics (deposit id: cfg_rRequest_3531).

## Discussion

*C. glabrata* possesses a large number of putative GPI-modified wall adhesins proposed to be important for the pathogenicity of this fungus. Based on homology in their effector domains these proteins were classified into seven different groups/clusters [1,5,7]. Members of the Epa cluster have been well-characterized, both functionally as well as structurally [8–11]. However, till now, the majority of the putative adhesins in other clusters lack characterization. Proteomic studies revealed that, besides members of the Epa cluster, cluster V and VI proteins, i.e. Awp1 —Awp4, can be commonly found in cell wall preparations of *C. glabrata* strains [4,5,12].

We showed by bioinformatic analysis that the N-terminal A-regions of cluster V and VI cell wall proteins belong to the same protein family and are highly related to other fungal proteins like the haze-protetective factor (Hpf1) from *Saccharomyces cerevisiae* [40] or hypha-specific GPI-CWPs from *Candida albicans* like Hyr1 or Iff4, members of the Iff protein family [1]. Interestingly, our SSN analysis found that these and other, related, fungal A-regions belong to a much larger cluster involving many bacterial orthologs (Fig 1). Structural analysis of the

Awp1 and Awp3b A-regions proved this relationship, as these A-regions correspond to a fusion between a right-handed parallel β-helix fold and an α-crystallin domain; Awp1-A and Awp3b-A are hence structurally very similar with a pairwise r.m.s.d of only 1.4 Å. Our Awp1-like A-domains are distinct from so far available structures of A-domains of *C. glabrata* adhesins, namely Epa1, Epa6, and Epa9, all belonging to adhesin cluster I. These Ca$^{2+}$-dependent adhesins contain a PA-14 domain that mediates glycan binding [8,9,41]. The Pwp family of *C. glabrata* adhesins, i.e. cluster II, is predicted to contain a PA-14 domain as well; however, no information for members of this family in terms of structures and cognate ligands is yet available [36].

The parallel β-helix of Awp1-related adhesins structurally resembles pectate lyases and pectate methyl transferases of fungal and bacterial origin, respectively, as all have eight complete β-turns [42,43]. However, there is no predictable, conserved active site found along the β-helix surfaces of Awp1-related adhesins. Conserved Ca$^{2+}$-binding sites, a prerequisite of enzymatic activity in polysaccharide lyases, are hence missing in structures of Awp1 and Awp3b A-regions. Furthermore, we failed to show by glycan arrays any interactions between these A-regions and a cognate oligosaccharide ligand. Besides other, glycan-dependent β-helix enzymes and tailspike proteins from bacteriophages, the Awp1-3 β-helices are structurally closely related to several bacterial secretion domains from autotransporter (AT) families (Fig 8A), some of them can form rather large β-helices. In pathogenic Gram-negative bacteria, these virulence factors are known to act as adhesins or haemolysins during host colonization, biofilm formation or adhesion to abiotic surfaces [44]. For example, a 500 nm long β-helix region that is located at the N-terminus of the filamentous hemagglutinin (FHA) from *Bordetella pertussis* sticks out of the outer membrane and apparently mediates a multitude of interactions, e.g. to host glycolipids, host proteins and abiotic surfaces [45]. Another cell-exposed protein, HxuA from *Haemophilus influenza* (Fig 8A), mediates binding to the host's serum

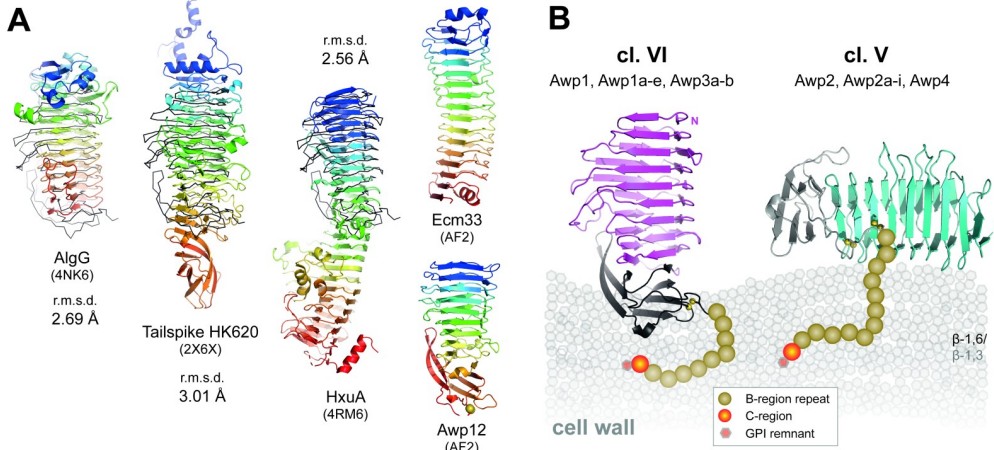

**Fig 8. Structural relationships and predicted arrangement on glycan cell wall of Awp1-related adhesins.** (A) Structural superposition of the Awp1 A-region (black ribbon) and three bacterial right-handed β-helix proteins colored from the N-terminus (blue) to the C-terminus (red): The periplasmic alginate epimerase AlgG from *Pseudomonas syringae*, the tailspike protein from the *E. coli* bacteriophage HK620 and the hemopexin binding protein HxuA from *Haemophilus influenzae*. The r.m.s.d. values after superposition were calculated for the respective β-helix domains by PDBeFold [27]. Right, Alphafold2 models of the Ecm33 domain from *S. cerevisiae* (Uniprot entry P38248, residues S28-S362) and Awp12 from *C. glabrata* CBS138 (Q6FSI9, D20-C314). (B) Predicted association modes of the A-regions of cluster V and VI adhesins to the cell wall of *C. glabrata*. The repetitive, highly O-glycosylated repeats of the B-region are schematically depicted as spheres (ochre). The C-region (red) of these *C. glabrata* GPI-CWPs links the cell wall proteins via a GPI-remnant (hexagon, orange) to the outer β-1,6-glucan layer (grey).

protein haemopexin for triggering bacterial iron uptake [46]. Given the topological similarities, we suggest that at least some of the Awp1-related adhesins may mediate specific interactions to host proteins, glycans or lipids as well. In addition, Awp1-related adhesins as analyzed in this study are apparently not the only fungal cell-wall anchored proteins with right-handed β-helices. For example, Alphafold2's template-free prediction [47] of Ecm33, a cell wall protein widely distributed in Ascomycetes, shows a β-helix with 13 turns (Fig 8A). This shows that β-helices apparently form a large group of ascomycetal cell wall proteins besides the well characterized adhesion domains of the PA-14 type [6,8,10,11,17,41]. In the context of *C. glabrata* CBS138, we found by Alphafold2 that adhesins from cluster III with its 13 members and cluster VII with six members [1] also adopt the A-region architecture of Awp1-related adhesins [48](Fig 8A). Surprisingly, this prompts the notion that the β-helix/α-crystallin family with its 42 assignable GPI-CWP (S5 Table) may play a similarly important role for the biology of *C. glabrata* as the PA-14 adhesion domain harboring GPI-CWP as the latter harbor only 20 members from the Epa [8–10] and 7 members from the Pwp-family [1,7].

The α-crystallin domain of Awp1-related adhesins apparently serves as a C-terminal cap for the β-helix fold, whose ninth turn is already incomplete with only two β-strands. *N*- and *O*-glycosylation sites are predicted to be rare in the β-helix fold, but numerous in the α-crystallin domain. We suggest that the glycosylated α-crystallin domain, alike the majority of the B-region, is embedded into the β-glucan portion of the *Candida* cell wall, where these Awp1-related adhesins are covalently linked via their GPI remnant at their C-terminal end. Cluster V adhesins like Awp2 but not cluster VI adhesins like Awp1 and Awp3b are predicted to cross-link the C-terminal CXXC motif with the central region of the β-helix region. The difference between cluster V and VI adhesins is then the presentation mode of the β-helix region, as for cluster VI members it is predicted to stick straight out, whereas cluster V adhesins place this elongated domain flat-on along the cell wall surface (Fig 8B).

Our phenotypic characterization of *awp1,3* and *awp2* deletion mutants demonstrated clearly a role of Awp2 but not Awp1 or the Awp3 paralogs in mediating hyperadhesiveness to polystyrene. For this, we deleted *AWP2* in three distinct genetic backgrounds of *C. glabrata*, i.e. reference strain CBS138 and the hyperadhesive isolates PEU382 and PEU427. Surprisingly, adherence to polystyrene was diminished only for the latter two but not the lowly adherent reference strain. Involvement of Awp2, but not Awp3, in adherence to polystyrene was further supported by studies with gain-of-function mutants expressing the A-domains of these putative adhesins on the yeast cell surface. Interestingly, consequences of deleting *AWP2* hugely depended on the genetic background. The *awp2Δ* mutants in PEU382 showed lowered adherence both after short (4 h) and long (24 h) incubation times. In contrast, strain PEU427, whose adherence to polystyrene is in-between PEU382 and reference strain CBS138, showed this behavior only for short incubation times (4 h). Furthermore, unlike the *awp2Δ* mutants in CBS138 and PEU427, the *awp2Δ* mutants in PEU382 also show a variety of other cell surface-related phenotypes, all of them converting the atypical surface features of PEU382 into a reference strain-like characteristics.

As CBS138, PEU382, and PEU427 all incorporate Awp2 in their cell walls the question is why the observed phenotypes are so distinct in these different genetic backgrounds. Based on proteomic peptide counts from biofilm cell wall preparations [12] (S2 Table), Awp2 is not more abundant in PEU382 than in the other two strains, therefore not providing an obvious explanation for its dominant role in PEU382. Another possiblity could be mutations in the *AWP2* gene in the different strains, but sequence analysis of the *AWP2* gene showed for both PEU strains, that originate from the same hospital, only a single $A_{2162}G$ mutation relative to CBS138. This mutation causes a $N_{721}S$ conversion in the C-terminal spacer domain of Awp2 and hence fails to provide a plausible explanation for phenotypical differences. Nevertheless, when analyzing the total peptide

counts of putative adhesins in proteomic data from biofilms of the three parental strains remarkable differences become apparent [12] (S2 Table). Compared to CBS138, both hyperadhesive strains show increased numbers of adhesin peptides. In PEU427, the dominant adhesin family appears to be cluster I, i.e. Epa-like adhesins, whereas in PEU382 cluster V, to which Awp2 belongs, is enriched. Thirteen genes of this cluster have been identified in the genome of CBS138. Therefore, we cannot exclude that specific regulation and functional redundancy complicate studies that aim to decipher the contribution or importance of an individual gene of such a large family. Apart of Awp2, all three strains harbor other cluster V adhesins in their cell walls. More specifically, Awp4 is present in all three strains like Awp2; Awp2a was found in both PEU strains, Awp2b and Awp2c in PEU427, and Awp2d is only present and abundant in PEU382. Future studies with mutants in other genes of the family, including deletions of multiple genes are therefore desirable to further functional characterization of these adhesins.

Our observation that Awp2 mediates *C. glabrata* adhesion to abiotic surfaces in some clinical isolates reminds of the function of other fungal cell wall proteins. For example, in the case of *C. albicans*, a phenotypical screen of 25 cell wall proteins for the ability to confer plastic adhesion identified Iff4, which additionally affects the infection burden in a mouse model for vaginal candidiasis [49]. Clearly, the N-terminal region of Iff4 is structurally related to Awp2 as it belongs to the family of Awp1-related adhesins as well (Fig 1). In *S. cerevisiae*, adhesion to hydrophobic surfaces like polystyrene as well as exceedingly stable cell-cell interactions are realized by Flo11 [26,50]. The adhesiveness of the Flo11 A-domain, at least to plastics, is mediated by two hydrophobic surface-exposed clusters of aromatic residues [18,26,51]. Interestingly, our model of the Awp2 A-region, but not the structures of Awp1 and Awp3b, shows a clustering of surface-exposed aromatic residues in the α-crystallin domain as well (S7 Fig). Together with the flat-on presentation mode of the A-region of Awp2 (Fig 8B), this suggests a distinct function for the α-crystallin domain, where the N-terminal β-helix domain may still fulfill other specificities in terms of adhesion or modulation of fungal cell wall characteristics. In this context one may even wonder whether some Awp1-related adhesins might confer an anti-adhesive function similar to Ywp1 from *C. albicans* [52]. Slightly reduced adhesiveness of Awp3b-overexpressing mutants was indeed observable (Fig 2D).

The impressive multitude of paralogous genes encoding for Awp1-related adhesins in *C. glabrata* or *C. albicans*, implies a certain degree of functional redundancy, which currently impedes clear-cut phenotypical assignments in *Candida* yeasts. Analogous to Epa-type adhesins, the batteries of GPI-CWPs with β-helix/α-crystallin modules most likely allow pathogenic yeasts to exploit different ecological niches in host organisms by modulating either their adhesive specificity or overall surface characteristics. An intriguing possibility is that some of the adhesins are involved in cell-cell interactions, e.g. for promoting binding to hyphae of *C. albicans* (Tati et al. 2016). This might allow *C. glabrata*, which strictly grows as a budding yeast, to enter host tissues, and is consistent with the notion that *C. glabrata* is often found in mixed infections with *C. albicans* causing oral candidiasis (Coco et al., 2008; Miranda-Cadena et al. 2018). Based on our results on Awp1- and Epa-like adhesins, future studies will enable us to understand how *C. glabrata* modifies its surface characteristics in response to different environmental surface cues.

## Materials & methods

### Sequence similarity network analysis of the A-domains of Awp1-related cell wall proteins

The A-regions of Awp1 (L19-R244, QHS67468.1) and Awp3a (G24-Q301, QHS67883.1) were used for iterative PSI-BLAST searches against the NCBI database of non-redundant protein

sequences. The cut-off criterion for the E-value was set to $10^{-4}$, the sequence coverage to at least 70%. After ten rounds, the aligned sequences were combined, redundant hits removed before subjecting the 11737 remaining sequences to a sequence similarity network (SSN) analysis with the Enzyme Similarity Tool of the Enzyme Function Initiative (EFI-EST)[53]. For initial SSN analysis, we used a BLAST-derived E-value stringency of $10^{-5}$. For edge removal, the alignment score E-value cut-off was decreased stepwise to finally $10^{-20}$ and $10^{-25}$, respectively. The resulting network with a pair-wise sequence identity greater than 80% for each node resulted in 4507 nodes. 10290 sequences were associated with full taxonomic description. The data were analyzed with Cytoscape 3.8.2, Clustal Omega and WEBLOGO [54–56].

## Generation of deletion mutants

Based on previous proteomic studies [12], *AWP2* deletion mutants were generated in reference strain CBS138 as well as in two hyperadherent strains (PEU427 and PEU382) [12], whereas mutants for cluster VI genes *AWP1 and AWP3* were generated in CBS138 only. An *awp1Δ/ awp3Δ* double mutant was also generated. The new version of the CBS138 genome [7] was published in the course of this work. The described 5 kb genomic insertion in the *AWP3* locus converts this gene in two tandem genes, *AWP3a* and *AWP3b*. Our *awp3Δ* and *awp1Δ/awp3Δ* mutants are lacking both *AWP3a* and *AWP3b*.

Deletion mutants were generated using the *SAT1*-flipper system [57] in combination with CRISPR-Cas9 to achieve more efficient integration into the correct genomic loci. The *SAT1*-flipper system, originally designed for *C. albicans*, harbours a *SAT1* gene for transformant selection and employs an inducible *FLP1* gene for excision of the integrated cassette from the genome to obtain clean mutants. To improve functionality of the flippase encoded in the deletion cassette in *C. glabrata*, the *C. albicans* adapted version (*CaFLP1*) of this gene in pSFS1a was replaced by the original *FLP1* gene from *S. cerevisiae*. For preparation of the *AWP* deletion cassettes, flanking regions of about 0.5 kb of the adhesin genes were amplified using proofreading KAPA polymerase and cloned into the *Kpn*I and *Xho*I (upstream fragment) and *Not*I and *Sac*I (downstream fragments) sites of the modified pSFS1a vector. Correctness of the deletion cassettes was verified by DNA sequencing. The RNA-protein (RNP) complex-based CRISPR-Cas9 system of IDT (*Leuven*, *Belgium*), whose application in *Candida* was described in detail by Grahl *et al.* (2017) [58], was employed for achieving correct integration into the genome of *C. glabrata*. Selection of transformants was performed on YPD (1% yeast extract, 2% peptone, 2% dextrose) agar containing 200 µg/mL nourseothricin (NT). Deletion mutants among transformants were identified by PCR. Correct integration in the genome was verified for both ends of the deletion cassettes using a combination of primers that covered the junctions, and deletion of the *AWP* genes was also verified with internal *AWP* gene primers. Deletion cassettes were removed from the genome by inducing the flippase gene, which is under control of the *CaSAP2* promoter, by growth in YCB-BSA-YE (1.17% yeast carbon base, 0.4% bovine serum albumin, 0.2% yeast extract, pH 4,0) and selecting slow-growing colonies on plates containing a low concentration of 10 µg/mL NT. Loss of the deletion cassettes was verified by PCR (Fig 2). For each *AWP* gene, at least two deletion mutants from independent transformation experiments were obtained and in all phenotypic assays data of mutants therefore represent the average of two mutants. Notably, the *AWP2* gene of the hyperadherent strains PEU382 and PEU427 harbors the point mutation N721S for the encoded adhesin as shown by sequencing. Oligonucleotides and gene-specific sgRNA guides used for this work are listed in S1 Table.

## Adhesion to polystyrene

Adhesion to polystyrene was measured using two different assays depending on the time allowed to adhere (4 or 24 h). For the 24 h experiment, overnight cultures in YPD at 37°C were adjusted to a cell density of $OD_{600} = 0.5$ in fresh YPD, and 200 μL of the cell suspension was pipetted into a 96-well plate and incubated at 37°C for 24 h in a humid environment. Unattached cells were removed by gentle washing with mQ water, and the remaining adhered cells were stained with 0.1% crystal violet (CV) solution for 30 min followed by washing with mQ water. Finally, CV was solubilized in 33% glacial acetic acid and quantified by measuring the $OD_{595}$ using a microplate reader (Molecular Devices). Data for each strain are the average of six technical and at least two biological replicates. Polystyrene adhesion of *S. cerevisiae* strains that overexpress hybrid *C. glabrata* constructs containing the A-domains of Awp2 and Awp3b (from BG2 background) at the cell surface, detailed in Zupancic et al., 2008 [8] and Tati et al., 2016 [59], was tested using the same assay but incubating in synthetic complete medium (2% glucose, 1.1% acid casein peptone, 0.6% ammonium sulfate, 0.2% yeast nitrogen base (YNB, without ammonium sulfate and without amino acids) at 30°C. Non-adhesive parental strain BY4741 and *C. glabrata* reference strain CBS138 were added as controls.

For the 4 h adhesion experiment, overnight cultures were diluted to $OD_{600} = 0.05$ in PBS, and 0.5 mL was pipetted in 12-wells plates or glass tubes and incubated for 4 h at 37°C. Unbound cells were removed by two washes with PBS after which adhered cells were treated for 10 min with a 2.5% trypsin (from porcine pancreas, *Sigma*) in PBS solution. Finally, PBS was added to a final volume of 0.5 mL, cells were resuspended and measured using a MACS-Quant (*Miltenyi Biotec*) flow cytometer (FC). Data are the average of two biological replicates measured in triplicate.

## Adhesion to HeLa and HaCaT cells

Confluent layers of HeLa and HaCaT cells in 12-well plates were prepared as described [60] but without antimycotics in the medium. Overnight cultures (YPD, 37°C) of *C. glabrata* were diluted in pre-warmed RPMI medium to obtain a cell density of about $6.0 \times 10^3$ cells/mL. Forty μl of the cell suspensions were added to the HeLa cells and incubated for 2 h at 37°C with 5% $CO_2$. Non-adhered cells were pipetted off. Adhered cells were scraped off with a cell scraper and collected in 2 mL PBS. Cells in both fractions were concentrated by centrifuging and plated on YPD plates containing 2 μg/mL chloramphenicol for colony-forming unit (CFU) counting after growth. For each sample, a control without HeLa cells was performed to discard adhesion to plastic. Data shown are averages of at least six biological replicates per sample.

## Drug sensitivity assays

Susceptibility to antifungals and cell wall perturbants was tested in 96-well plates following EUCAST guidelines. Twofold serial dilutions of compounds were prepared in YPD and mixed 1:1 with cells from overnight cultures diluted to an $OD_{600}$ of 0.01. Plates were incubated for 24 h at 37°C. Minimal inhibitory concentrations (MIC) were determined after reading the $OD_{600}$ in a microplate reader. Compounds tested were amphotericin B, fluconazole, isavuconazole, caspofungin, micafungin, Calcofluor white and SDS. For CFW drop assays were done as an alternative assay for sensitivity. Tenfold serial dilutions were prepared from overnight cultures, and 4 μl were spotted on YPD plates containing 200 μg/mL CFW. Growth was monitored after 24 and 48 h of growth at 37°C.

### Growth rate and sensitivity to zymolyase

For determination of growth rates, cells from overnight cultures were inoculated 1/100 in fresh YPD and incubated in 200 µl volumes in 96-well plates at 37˚C with agitation in a SpectraMax 340PC microplate reader. The increase in $OD_{600}$ was followed in time. For determination of zymolyase sensitivity, cells were grown in YPD until log phase, collected, and resuspended in 10 mM Tris-HCl, pH 7.5 at an $OD_{600}$ of 2.0 to which 0.25% of β-mercaptoethanol was added. After 1 h of incubation at room temperature (RT), 180 µl of cell suspensions and 20 µl of 10 U/mL zymolyase were mixed in a 96-well plate and placed in the microplate reader at 37˚C. Decrease in $OD_{600}$ was measured each minute after a short mixing pulse. Curves represent averages of three technical and two biological replicates.

### Cell surface hydrophobicity

Exponential phase and overnight cultures in YPD at 37˚C were washed two times with PBS and resuspended at an $OD_{600}$ of 0.7. Cell suspensions were mixed with hexadecane in glass tubes at a 15:1 volume ratio. Upon 1 min of gentle vortexing, the phases were allowed to settle for 10 min after which the $OD_{600}$ of the aquous phase was measured. Each strain was assayed twice with two technical replicates each.

### Sedimentation and aggregation

Overnight cultures in YPD at 37˚C were transferred to glass tubes. Every 10 min 50 µl of sample was carefully taken 2 cm below the surface of the cell suspension for OD600 measurements. After 1 h the cells were washed with PBS and the sedimentation experiment was repeated. The sedimentation test with PEU382 was also performed after adjusting the pH of the cell culture to 6.5 with PBS or NaOH.

Cell aggregation of overnight cultures in YPD was observed with a Leica DM1000 microscope mounted with a MC170 HD digital camera.

### Cell wall protein and chitin content

Cells grown to logarithmic phase in YPD at 37˚C were harvested or seeded in Petri dishes for biofilm development as detailed in Gómez-Molero *et al.* [12]. Preparation of cell walls was performed using a Fastprep 24 instrument (*MP Biomedicals*) as described in Gómez-Molero *et al.* [12] and references therein. Protein and chitin content were determined using colorimetric assays as described by Kapteyn *et al.* [61].

### Statistical analysis

Statistical significance of phenotypic data was analyzed by Student's t-tests or one-way ANOVA followed by post hoc Delayed Matching to Sample (DMS) tests. P values <0.05 were considered statistically significant.

### Cloning of AWP1 and AWP3b in expression plasmids

N-terminal effector domains of Awp1 and Awp3b were amplified using KAPA polymerase (Kapa Biosystems) and cloned into the *Bam*HI and *Hin*dIII restriction sites of expression plasmid pET28a (Novagen). Correctness of the clones was verified by DNA sequencing (*STAB-Vida*). Expression was performed in *E. coli* SHuffle T7 Express cells (*New England Biolabs*). Cells were grown at 37˚C in Luria broth (LB) medium to an $OD_{600}$ of 0.6. Overexpression was induced by addition of 0.1 mM isopropyl-β-D-thiogalactopyranoside and 72 h further incubation at 12˚C and 140 rpm in baffled flasks. Cells were harvested by centrifugation at 3200 g,

4˚C, 20 min, washed with 50 mM $NaH_2PO_4$, 300 mM NaCl, pH 8.0 buffer, and stored at -80˚C until use.

## Protein purification

Cell pellets were resuspended in Ni-NTA buffer and lysed using a microfluidizer (*Microfluidics*). Cell lysate was cleared by centrifugation at 18000 rpm, 4˚C, for 40 min. Supernatant was filtered using a 0.45 μm syringe filter (*Millipore*) and applied on a 5 mL Ni-NTA column (*Macherey Nagel*), equilibrated with Ni-NTA buffer. Protein was eluted stepwise using increasing imidazole concentrations; eluted fractions were analyzed by SDS-PAGE. Fractions containing the target protein were pooled and concentrated using an Amicon Ultra concentrator with 30 kDa cut-off (*Millipore*). Concentrated protein samples were further purified by size exclusion chromatography, on a Superdex 200 pg column (*GE Healthcare*), equilibrated with SEC buffer (20 mM Tris-HCl, 300 mM NaCl, pH 8.0). All purification steps were performed at a temperature of 4˚C. Finally, protein purity was assessed by SDS-PAGE; protein solutions were stored at 4˚C.

## Protein crystallization

The Awp1 A-region was crystallized in a hanging-drop vapor diffusion setup in a 1.2 μL drop (0.6 μL protein solution + 0.6 μL reservoir), equilibrated against 1 mL 0.1 M MOPSO/bis-tris pH 6.5, 10% (w/v) PEG 8000, 20% 1,5-pentanediol, 0.5 mM erbium (III) chloride hexahydrate, 0.5 mM terbium (III) chloride hexahydrate, and 0.5 mM ytterbium (III) chloride hexahydrate at a temperature of 18˚C. A protein concentration of 48 mg/mL was used, crystals were grown for 3–4 weeks. The Awp3b A-region was crystallized at a protein concentration of 24 mg/mL in a hanging-drop vapor diffusion setup in a 0.6 μL drop (0.3 μL protein solution + 0.3 μL reservoir), equilibrated against 1 mL of 0.2 M magnesium chloride, 0.1 M Tris pH 7.0, 3.0 M sodium chloride at an incubation temperature of 18˚C. Crystals were allowed to grow for 2–3 weeks. Crystals used for collection of the native dataset of Awp3b A-region were grown in a sitting drop vapor diffusion setup. A 0.6 μL drop (0.3 μL protein solution + 0.3 μL reservoir) was equilibrated against 50 μL 0.1 M sodium phosphate, 0.1 M potassium phosphate, 0.1 M MES pH 6.5, 2.0 M sodium chloride. Crystals were harvested after several months.

## Data collection and processing

For collection of native datasets, crystals were harvested and directly flash-frozen in liquid nitrogen without additional cryoprotection. To enable phase determination of the Awp3b A-domain by single wavelength anomalous diffraction (SAD), crystals were transferred to a drop of mother liquid, containing 50 mM gadolinium (III) acetate. Crystals were allowed to sit in the drop for 90 min, and then flash-frozen in liquid nitrogen without additional cryoprotection. Datasets were collected at 100 K at the European Synchrotron Radiation Facility (ESRF), Grenoble. Diffraction images were integrated in *iMOSFLM* [62] or *XDS* [63], data reduction was done in *AIMLESS* [64], run in the *CCP4i2* [65] software suite.

## Structure solution and refinement

Crystallographic phases of the Awp3b-$Gd^{3+}$ complex were determined using *CRANK2* [66], followed by refinement in *REFMAC5* [67] and model building in *ARP/wARP* [68]. The $Gd^{3+}$-derivative was used to solve a native dataset of the Awp3b A A-region with higher resolution. The phases of the Awp1 A-region were determined via molecular replacement in *Phaser* [69], using the structure of Awp3b A-region as a starting model, followed by 20 cycles of

structure building using the *ARP/wARP Web Service* [68]. All structures were refined running iterative cycles model building in *Coot* [70] and *phenix.refine* [71] or *REFMAC5* [67]. A summary of the data collection and refinement statistics is given in Table 2.

## Supporting information

**S1 Table. Oligonucleotides and sgRNA guides used in this study.**
(XLSX)

**S2 Table. Semi-quantitative analysis of proteomics data of biofilm wall samples from Gomez-Molero et al. 2015 [12] based on peptide counting.**
(XLSX)

**S3 Table. Drug sensitivity of *AWP* mutants.**
(XLSX)

**S4 Table. *O*- and *N*-glycosylation sites of cluster V/VI Awp1-related adhesins of *C. glabrata* as predicted by NetOGlyc4.0.** For comparison the prediction is also shown for the haze-protective factor, Hpf1, of *Saccharomyces cerevisiae*. Notably, the Awp1-related region of Hpf1 (S337-S629) is not located at the N-terminus, but flanked by highly O-glycosylated repeat regions.
(XLSX)

**S5 Table. Other Awp1-related adhesins in *C.glabrata* with right-handed β-helix domain as predicted by Alphafold 2.** Besides cluster III and VII adhesins, three additional have been found by our SSN analysis (Fig 1). The open reading frame CAGL0J05159g of *C. glabrata* CBS138 (XP_002999571.1) has not been assigned to a gene entry in the assembly of Xu et al. 2020, due to a frameshift that is not found in other strains. However the N-terminal region also adopts the fold of an Awp1-related adhesin.
(XLSX)

**S1 Fig. Structure-based multiple sequence alignment (MSA) of Awp1-like GPI-CWPs of *C. glabrata*.** The MSA was generated for A-regions of *C. glabrata* GPI-CWPs with β-helix domains by 3D-coffee using the structural information of the Awp1-A and Awp3b-A domains. The nine β-helix turns are highlighted by blue-to-yellow coloured boxes; the α-crystalline domain by a cyan box. Cysteines are highlighted in yellow. Notably, pairwise sequence identities of Awp2, Awp2a-Awp2i and Awp4 A-regions vary from 53.6% to 94.8% (cluster V), those of Awp1,Awp1a-Awp1e, Awp3a and Awp3b (cluster VI) from 18.1% to 64.1%. For clarity, distantly related Awp1-like adhesins of cluster V (QHS65613.1, QHS67215.1, QHS68879.1) are omitted from the MSA. Anyway, pairwise sequence identities between clusters V and VI A-regions are low, i.e. in the range of 14.4% to 22.9%.
(TIF)

**S2 Fig. *AWP* gene deletion strategy and PCR confirmation of deletion mutants.** (A) Schematic structure of *SAT1*-flipping deletion constructs and *AWP* gene deletion procedure. CRISPR-Cas9 RNP complexes (not indicated) are added during transformation to aid integration into the correct locus. Mutants are selected by PCR analysis of 5' and 3' integration junctions, after which the cassette is excised by inducing *FLP1* expression. (B-D). PCR verification of *AWP* deletion mutants after excision of the cassette using external (Ext) and internal (Int) primers as indicated in (A). (B) Verification of *awp1Δ* mutants. (C) Verification of *awp3abΔ* mutants and *awp1Δ /awp3abΔ* mutants. (D) Verification of *awp2Δ* mutants in three different

genetic backgrounds.
(TIF)

**S3 Fig. Awp2-mediated adhesion to plastics is metal-independent.** Adhesion to polystyrene after 24 h of incubation in YPD in the presence of the indicated EDTA concentrations. Notably, the $MIC_{50}$ for all four tested strains is 0.8 mM.
(TIF)

**S4 Fig. Protein and chitin content.** Protein and chitin amount in the cell wall was measured using colorimetric assays as described in Kapteyn et al. (2001). Each strain was assayed twice with two technical replicates each. Statistically significant differences (Student's t-tests or ANOVA, $p < 0.05$) of mutants compared to their parental strain under the same conditions are indicated by asterisks.
(TIF)

**S5 Fig. Sedimentation and aggregation of *C. glabrata* strain PEU382.** (A) Sedimentation of a 37°C overnight culture of hyperadhesive aggregating strain PEU382 in YPD with and without pH adjustment by adding PBS (final conc. 1×) or NaOH was followed in time. (B) Sedimentation of PEU382 cells grown overnight at 37°C in YPD without or with supplementations as indicated. (C) Microscopical images of cells from (B) at t = 0.
(TIF)

**S6 Fig. *A priori* predicted contact maps of Awp1-related A-regions.** Contact maps were predicted by trRosetta [19] using therewith derived multiple sequence alignments of 532, 195 and 140 homologous sequences for Awp1, Awp2, and Awp3b, respectively. The TM-scores of the obtained models were very high with values of 0.738–0.839. Predicted contacts between the C-terminal end of the Awp2 A-region and β-helix turns 7 and 8 are highlighted by arrows. These contacts are the base for the disulfide bridges C327-C201 and C330-C184 in the Awp2 model.
(TIF)

**S7 Fig. Aromatic residues exposed on the surfaces of Awp1-related A-regions.** Only tyrosines have been found to be surface-exposed on Awp1-related A-regions. Awp2 harbors six residues each in the β-helix (coloured) and α-crystallin (gray) domain.
(TIF)

## Acknowledgments

We thank Diana Kruhl (Philipps-Universität Marburg), Ana Moreno and Albert de Boer (UCLM, Molecular Mycology), Diego Fernández (UCLM, Molecular Oncology), and the staff of Beamline 14.1 at the BESSYII synchrotron in Berlin (Germany) and Beamline ID23-1 at the European Synchrotron Radiation Facility (ESRF) in Grenoble (France) for technical support. David F. Smith and Jamie Heimburg-Molinaro of the Consortium for Functional Glycomics (CFG) are thanked for performing *in vitro* screening of glycan specificity. Brendan Cormack (Baltimore, MD) is thanked for kindly providing *S. cerevisiae AWP* overexpression plasmids, and Oliver Bader (Göttingen, Germany) for providing PEU strains.

## Author Contributions

**Conceptualization:** Piet de Groot, Lars-Oliver Essen.

**Data curation:** Viktoria Reithofer.

**Formal analysis:** Lars-Oliver Essen.

**Funding acquisition:** Piet de Groot, Lars-Oliver Essen.

**Investigation:** Viktoria Reithofer, Jordan Fernández-Pereira, María Alvarado.

**Methodology:** Piet de Groot, Lars-Oliver Essen.

**Project administration:** Lars-Oliver Essen.

**Resources:** Piet de Groot.

**Supervision:** Piet de Groot, Lars-Oliver Essen.

**Validation:** Lars-Oliver Essen.

**Visualization:** Viktoria Reithofer, María Alvarado, Piet de Groot.

**Writing – original draft:** Viktoria Reithofer, Jordan Fernández-Pereira, Piet de Groot, Lars-Oliver Essen.

**Writing – review & editing:** Piet de Groot, Lars-Oliver Essen.

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
