## [Decision Letter · Decision Letter 0]

15 Oct 2021

Dear Dr Essen,

Thank you very much for submitting your manuscript "A novel class of Candida glabrata cell wall proteins with β-helix fold mediates adhesion in clinical isolates" for consideration at PLOS Pathogens. As with all papers reviewed by the journal, your manuscript was reviewed by members of the editorial board and by several independent reviewers. In light of the reviews (below this email), we would like to invite the resubmission of a significantly-revised version that takes into account the reviewers' comments.

Thank you very much for submitting your manuscript "A novel class of Candida glabrata cell wall proteins with beta helix fold mediates adhesion in clinical isolates (PPATHOGENS-D-21-01943) for review by PLoS Pathogens. Your manuscript was fully evaluated at the editorial level and by independent peer reviewers. The reviewers appreciated the attention to an important problem, but raised some substantial concerns about the manuscript as it currently stands. These issues must be addressed before we would be willing to consider a revised version of your study. We cannot, of course, promise publication at that time. We therefore ask you to modify the manuscript according to the review recommendations before we can consider your manuscript for acceptance. Your revisions should address the specific points made by each reviewer.

I am returning your manuscript with three reviews. The reviewers varied in their assessment of the paper, as you will see. After reading the reviews and looking at the manuscript, I recommend Major Revision based on the critiques from the more critical reviews. I am sorry I cannot be more positive at this time, however we are looking forward to receiving your revision. With some effort, the manuscript will be suitable for a resubmission, if you so wish to do so.

Reviewers 1 and 2 raised concerns about the metal binding properties. Reviewer 1 in particular about the interpretation of Gd3+, the approaches to identifying the metal binding regions, the glycan studies, and the binding to disparate surfaces. Reviewer 3 pointed out hat the HeLa cells widely differ and perhaps are not the most appropriate cell to rely on for these studies. Perhaps, use of another cell line or primary cells would complement the data obtained with HeLa cells. Reviewer 2 suggested that additional experiments be conducted that relate to the conditions you have used. Reviewer 2 also had a number of issues with the writing. While these are useful to improve the quality of the manuscript they did not factor into the overall assessment.

Note that we may send your paper back to some of the more critical reviewers upon resubmission. Please pay particular attention to the following reviewer suggestions and give them due consideration. You must provide data on the nature and effectiveness of 'other effective methods' for reconsideration. Please include a table of these other methods for easy reference. In addition, when you are ready to resubmit, please be prepared to provide the following:

(1) A letter containing a detailed list of your responses to the review comments and a description of the changes you have made in the manuscript.

(2) Two versions of the manuscript: one with either highlights or tracked changes denoting where the text has been changed; the other a clean version (uploaded as the manuscript file). We hope to receive your revised manuscript within 60 days. If you anticipate any delay in its return, we ask that you let us know the expected resubmission date by replying to this email. Revised manuscripts received beyond 60 days may require evaluation and peer review similar to that applied to newly submitted manuscripts

We cannot make any decision about publication until we have seen the revised manuscript and your response to the reviewers' comments. Your revised manuscript is also likely to be sent to reviewers for further evaluation.

Sincerely,

George Deepe, jr., MD

Guest Editor

PLOS Pathogens

Sarah Gaffen

Section Editor

PLOS Pathogens

Kasturi Haldar

Editor-in-Chief

PLOS Pathogens

orcid.org/0000-0001-5065-158X

Michael Malim

Editor-in-Chief

PLOS Pathogens

orcid.org/0000-0002-7699-2064

Thank you very much for submitting your manuscript "A novel class of Candida glabrata cell wall proteins with beta helix fold mediates adhesion in clinical isolates (MS # PPATHOGENS-D-21-01943) for review by PLoS Pathogens. Your manuscript was fully evaluated at the editorial level and by independent peer reviewers. The reviewers appreciated the attention to an important problem, but raised some substantial concerns about the manuscript as it currently stands. These issues must be addressed before we would be willing to consider a revised version of your study. We cannot, of course, promise publication at that time. We therefore ask you to modify the manuscript according to the review recommendations before we can consider your manuscript for acceptance. Your revisions should address the specific points made by each reviewer.

I am returning your manuscript with three reviews. The reviewers varied in their assessment of the paper, as you will see. After reading the reviews and looking at the manuscript, I recommend Major Revision based on the critiques from the more critical reviews. I am sorry I cannot be more positive at this time, however we are looking forward to receiving your revision. With some effort, the manuscript will be suitable for a resubmission, if you so wish to do so.

Reviewers 1 and 2 raised concerns about the metal binding properties. Reviewer 1 in particular about the interpretation of Gd3+, the approaches to identifying the metal binding regions, the glycan studies, and the binding to disparate surfaces. Reviewer 3 pointed out hat the HeLa cells widely differ and perhaps are not the most appropriate cell to rely on for these studies. Perhaps, use of another cell line or primary cells would complement the data obtained with HeLa cells. Reviewer 2 suggested that additional experiments be conducted that relate to the conditions you have used. Reviewer 2 also had a number of issues with the writing. While these are useful to improve the quality of the manuscript they did not factor into the overall assessment.

Note that we may send your paper back to some of the more critical reviewers upon resubmission. Please pay particular attention to the following reviewer suggestions and give them due consideration. You must provide data on the nature and effectiveness of 'other effective methods' for reconsideration. Please include a table of these other methods for easy reference. In addition, when you are ready to resubmit, please be prepared to provide the following:

(1) A letter containing a detailed list of your responses to the review comments and a description of the changes you have made in the manuscript.

(2) Two versions of the manuscript: one with either highlights or tracked changes denoting where the text has been changed; the other a clean version (uploaded as the manuscript file). We hope to receive your revised manuscript within 60 days. If you anticipate any delay in its return, we ask that you let us know the expected resubmission date by replying to this email. Revised manuscripts received beyond 60 days may require evaluation and peer review similar to that applied to newly submitted manuscripts

Reviewer's Responses to Questions

**Part I - Summary**

Reviewer #1: This is an interesting manuscript that moves the field forward by providing first functional and structural characterization of three adhesins from two phylogenetic clusters of Candida glabrata GPI-anchored cell wall proteins (GPI-CWPs). The authors used bioinformatic approaches to classify the C. glabrata GPI-CWPs into 7 clusters based on sequence similarity; of these clusters, only adhesins in cluster I have been well characterized. The current manuscript presents the crystal structures of two adhesins from cluster V and a structural model for a cluster VI adhesin, along with a number of biological assays conducted with WT, knockout, and gain-of-function strains, including adhesion to polystyrene and HeLa cells; drug sensitivity; and growth, sedimentation, and aggregation assays. The structures of Awp1 and Awp3b A-domains reveal a β-helix and crystallin-like β-sandwich domain architecture that is reminiscent of pectate lyases and other bacterial glycan-binding proteins. There are a number of interesting structural features that may have biological relevance, although in a few cases these possibilities were not fully supported by biological or biochemical data (or attempts to do so were made, but with inconclusive results). The manuscript is very well written, with clear and informative figures. The experiments in general were conducted appropriately with good methodological descriptions, although as described below, there are a few additional experiments that would strengthen the manuscript.

Reviewer #2: This manuscript describes a novel fungal gene family and the first structures from that family for adhesins in Candida glabrata. These proteins have an N-terminal �-helical domains. The sequence analyses and the structures reported are comprehensively performed and by-and-large well described. There are homologs in other fungal pathogens as well. Therefore, the work is novel and highly important in fungal pathogenesis. However, a number of matters should be clarified and resolved before the paper is publishable. The most serious issues are clarification of the putative cell wall attachment mode and a discussion of the role of cations in binding.

Reviewer #3: The manuscript “A novel class of Candida glabrata cell wall proteins with β-helix fold mediates adhesion in clinical isolates” by Reithofer et al. utilized sequence similarity network analysis to study cluster V and VI cell wall adhesions of Candida glabrata. The authors use protein crystallization to show that the A-domains from cluster VI proteins Awp1 and Awp3b have a β-helix domain. Further analysis revealed that one of these uncharacterized cell wall proteins, namely Awp2 mediates adhesion to plastic in hyperadhesive clinical isolates.

**Part II – Major Issues: Key Experiments Required for Acceptance**

Reviewer #1: There are a few questions or concerns that should be addressed:

• One of my major questions had to do with the possible role of metal cations in the function of these adhesins (particularly Awp3b). The Gd3+-soaked crystals used for phase determination showed an impressive array of Gd3+ ions, most of which were not directly coordinated by surface Awp3b residues. However, several of these Gd3+ ions were surface-bound; given the common use of lanthanides to replace Ca2+ ions in specific sites on proteins, it is an intriguing finding. I suspect the surface-bound Gd3+ positions may represent low-affinity metal binding sites for Ca2+ or another metal cation.

• In terms of the potential metal binding sites on Awp3b, were any computational approaches used to identify likely metal binding sites in the structure? There are various programs or servers mentioned in the literature for this type of analysis such as CheckMyMetal (http://csgid.org/csgid/metal_sites/). It would be a good idea to use this server or a similar program to see whether the surface-bound Gd3+ are consistent with a bona fide binding site for Ca2+ or another metal ion. Was any electron density visible in the native Awp3b (7O9O) structure at the locations where Gd3+ ions were observed at the protein surface in the phased structure (7O9Q)?

• The authors mention at the top of page 5 that washing (or diluting) the PEU382 cells with PBS abolished hyperaggregation and typical sedimentation behavior, which could potentially be due to the loss of surface-bound Ca2+ (or other) metal ions, resulting in loss of intercellular adhesion. Although dilution with NaOH instead of PBS failed to show the same effect, phosphate is known to cause solubility issues with many divalent cations such as Ca2+, so the PBS effect could still be responsible for metal loss. Therefore, I would strongly recommend conducting hyperaggregation, sedimentation, and adhesion assays by adding a strong chelator such as DTPA or EDTA to remove any bioavailable metal cations to see if the adhesive functions are lost; then various metal cations can be added back to a minimal inhibitory concentration of chelator to check for restoration of the adhesive function. These experiments can also be done using chemically defined media, but the chelator experiment is usually technically more feasible. Even low-affinity metal binding sites could certainly modulate the adhesive properties of these Awp proteins.

• The authors mentioned that the proteins were submitted for glycan array screening but that no glycan ligands were identified. What buffer was used for the glycan array screening? If there is a requirement for metal ions for adhesive function (as is often the case with protein-glycan interactions), potential ligands could be missed using a minimal buffer without the necessary metals.

• The exterior stacking of residues on the surface of the Awp structures is very interesting and reminiscent of bacterial β-helical ice-binding proteins, which form a flat planar protein surface that will form periodic H-bonds with the surface of ice crystals (see for example, Mangiagalli et al., doi: 10.1111/febs.14434; or Guo et al., doi: 10.1126/sciadv.1701440). Although this would not be relevant to the binding of hydrophobic polystyrene surfaces, it might be relevant to adhesion to other abiotic hydrophilic surfaces such as glass. Does C. glabrata adhere to both polystyrene and glass surfaces? Do the residues in these exterior stacking interactions form a flat protein surface that might interact with a planar abiotic substrate? Consistent with this idea, it’s interesting that very few (or zero) predicted O-glycosylation sites were identified in the β-helix subdomains, given that glycans would sterically inhibit interaction with an abiotic surface.

Reviewer #2: Gd++ binding and the effects of a PBS wash strongly suggest cation binding is key to activity. This point needs to be clarified throughout. PBS was used in the 4 hr adhesions, but not the 24 hr adhesions. Thus, effects of PBS, neutral pH and cation binding in Awp2-expressing cells (and also Awp3-expressing ?) are critical questions. Adhesion experiments at neutral pH w/wo EDTA and added cations would help point us in the right direction.

Reviewer #3: The authors used Hela cells to determine the contribution to adhesion to human cells. The gene and protein expression of Hela cells vary from lab to lab (PMID: 30778230). Furthermore, these cells are derived from a cancerous cervical tumor and are not reflective of a normal barrier cell (epithelial or endothelial cell). Given that the authors should consider primary human cells.

**Part III – Minor Issues: Editorial and Data Presentation Modifications**

Reviewer #1: • On page 4, 2nd line of final paragraph: the abbreviations CFW, CR, and SDS are used without prior explanation. Although SDS is fairly standard, CFW and CR are not and should be spelled out the first time they are used.

• In the final paragraph on page 6, a series of residues that form a Ser/Thr ladder are described, with a reference to Figure 6 (“…S45, S65, T92, T120, T148…”). It would be very helpful to label (at least some of) these residues in Fig 6, since the residues forming the outer stack in the +2 residue position are already labeled and it’s confusing to then look through all the labels and not see those listed for the Ser/Thr ladder.

• The PDB validation reports listed several RSRZ outliers in a couple of the structures, but this analysis is known to be rather unreliable by the crystallography community. To get a better validation report, please run the structures through the MolProbity server and report the ClashScore and percentiles in Table 1. (Other than the questionable RSRZ values, the crystal structures appeared to be well refined).

Reviewer #2: Abstract

1. Define ‘clusters’: are these homologous clusters, clusters of chromosomal loci, or both?

2. “Awp3b-A” meaning is not clear. Should it be “The A domain of Awp3b (Awp3b-A)…”?

3. Why is Gd binding important? Please see #16 below on cation binding.

4. “…was observed and phenotypic...” is run-on. I suggest “…was observed. Phenotypic...”

5. “…for adhesiveness, as well as..” I suggest “…for adhesion to polystyrene, as well as…”

6. The sentence “In contrast, deletion…” is long and hard to read.

Intro

p.2, para 1

7. Define ‘clusters’ Are there physically linked clusters, or just sequence-similar? See #1

8. “Although phylogenetically being more

9. “..effector domains, or functional domains –“

10. “By being linked with its processed C-terminus to cell wall β-1,6-glucans via the glycan remnant of the GPI anchor, the heavily glycosylated B-region can interact with other cell wall glycans and act as spacer molecule to breach through the wall to present the A-region along the cell surface (1, 6).” Here and Fig. 8: it is likely that the modified GPI’s are linked to the outer layers of wall glucan, and so may not need to ‘breach through the wall’.

11. Para 2 “in other fungi, which may raise”[suggest “a finding which raises”]

12. Para 3 planktonic growth (4),” specify which strain

13. Para 5 “Phylogenetic analysis revealed weak structural relationships” [suggest “Sequence analysis revealed weak similarities leading to predictions of structural similarities...”

Phylogenetic analysis revealed weak structural relationships

14. Para 1 “…S. cerevisiae harbors only two, Hpf1 and Css1.” S. cerevisiae Awa1 mentioned is mentioned later as a homolog, but is not listed here

15. Para 2 In Candida glabrata CBS138 we find [found] 22

Awp2 but not Awp1 or Awp3 governs adhesion to polystyrene

16. Gd++ binding and the effects of a PBS wash strongly suggest cation binding is key to activity. This point needs to be clarified throughout. PBS was used in the 4 hr adhesions, but not the 24 hr adhesions. Thus, effects of PBS, neutral pH and cation binding in Awp2-expressing cells (and also Awp3-expressing ?) are critical questions. Adhesion experiments at neutral pH w/wo EDTA and added cations would help point us in the right direction.

17. “…PEU427 and CBS138 and awp1,3Δ double mutants…” should be “..PEU427 and CBS138, because awp1,3Δ…”

The two-domain architecture of Awp1-related A-regions

18. Para 2 “The roles of the Ser/Thr and the Asn stacks on the surfaces Awp1-related A-domain β-helices hence remain elusive.” See Richardson, J. S. (1981) The anatomy and taxonomy of protein structure. Adv. Protein Chem. Biochem. 35, 167–339. This review clearly predicts the roles of ASN in the �-helical turns.

Structural relationship to other β-helix proteins

19. “RMSD values ranging from 2.63 to 6.04 Å” --In this reviewer’s opinion, 6 Å is a characteristic of structural similarity, but not ‘high structural similarity’

20. The analyses seem to predict that Awp2 bind divalent cations and glycans. Cellular adhesion assays with EDTA and divalents will clarify this point. See #3, #16

Discussion

21. Para 4. The discussion of cell wall association implies membrane binding and penetration through the wall. This is probably incorrect, according to previous publications by one of the authors. See doi: 10.1128/EC.00284-08.

22. Para 7. “residues (18, 25)” There is a recent highly relevant reference: doi: 10.7554/eLife.6859

M&M

Generation of deletion mutants

23. line 2: (12)_ whereas

Legends

24. Fig. 1: Title: should be “SSN analysis…

25. The edge inclusion criteria in the L and R graphics should be specified.

26. Add an explanantion of the coloring, e.g.: “The nodes are colored by fungal clade, with Red being C. glabrata (Nakaseomyces). Light orange designated the CTG clade, including Candida albicans.”

Fig. 2

27. Define FC

28. Correct to: “Data in (B)[A] and (C) are normalized to CBS138…”

Fig. 3

29. 3D: Add “200 ug/ml calcofluor white” to Y axis label

30. 3F top panels need better contrast

31. Fig. 6; Awp 2 is not shown and should be omitted from the title

32. Fig. 8 Alter the CW attachment cartoon to acknowledge attachment to glucan. See #10 and #21.

Reviewer #3: The authors used a static assay to determine adhesion to polystyrene showing that Awp2 mediates binding to plastic in a time dependent manner. It is know that flow conditions might change binding of fungi to plastic as seen in catheters. Therefore, it would be interesting to see the contribution of Awp2 to adhesion under flow conditions.

PLOS authors have the option to publish the peer review history of their article (what does this mean?). If published, this will include your full peer review and any attached files.

Reviewer #1: **Yes: **Andrew B. Herr

Reviewer #2: No

Reviewer #3: No
---

## [Editor Report · Decision Letter 1]

30 Nov 2021

Dear Dr. Essen,

We are pleased to inform you that your manuscript 'A novel class of Candida glabrata cell wall proteins with β-helix fold mediates adhesion in clinical isolates' has been provisionally accepted for publication in PLOS Pathogens.

Best regards,

Sarah L. Gaffen, PhD

Section Editor

PLOS Pathogens

Sarah Gaffen

Section Editor

PLOS Pathogens

Kasturi Haldar

Editor-in-Chief

PLOS Pathogens

orcid.org/0000-0001-5065-158X

Michael Malim

Editor-in-Chief

PLOS Pathogens

orcid.org/0000-0002-7699-2064

The authors have satisfied the issues raised by the reviewers.
---

## [Editor Report · Acceptance letter]

15 Dec 2021

Dear Dr. Essen,

We are delighted to inform you that your manuscript, "A novel class of Candida glabrata cell wall proteins with β-helix fold mediates adhesion in clinical isolates," has been formally accepted for publication in PLOS Pathogens.

Best regards,

Kasturi Haldar

Editor-in-Chief

PLOS Pathogens

orcid.org/0000-0001-5065-158X

Michael Malim

Editor-in-Chief

PLOS Pathogens

orcid.org/0000-0002-7699-2064